# Phytochemicals and Their Correlation with Molecular Data in *Micromeria* and *Clinopodium* (Lamiaceae) Taxa

**DOI:** 10.3390/plants11233407

**Published:** 2022-12-06

**Authors:** Dario Kremer, Valerija Dunkić, Ivan Radosavljević, Faruk Bogunić, Daniella Ivanova, Dalibor Ballian, Danijela Stešević, Vlado Matevski, Vladimir Ranđelović, Eleni Eleftheriadou, Zlatko Šatović, Zlatko Liber

**Affiliations:** 1Faculty of Pharmacy and Biochemistry, University of Zagreb, A. Kovačića 1, 10000 Zagreb, Croatia; 2Faculty of Science, University of Split, Rudera Boškovića 33, 21000 Split, Croatia; 3Faculty of Science, University of Zagreb, Marulićev trg 9A, 10000 Zagreb, Croatia; 4Centre of Excellence for Biodiversity and Molecular Plant Breeding, Svetošimunska cesta 25, 10000 Zagreb, Croatia; 5Faculty of Forestry, University of Sarajevo, Zagrebačka 20, 71000 Sarajevo, Bosnia and Herzegovina; 6Institute of Biodiversity and Ecosystem Research, Bulgarian Academy of Sciences, Acad. Georgi Bonchev Str., bl. 23, 1113 Sofia, Bulgaria; 7Slovenian Forestry Institute, Večna Pot 2, 1000 Ljubljana, Slovenia; 8Faculty of Natural Sciences and Mathematics, University of Montenegro, Džordža Vašingtona bb, 81000 Podgorica, Montenegro; 9Faculty of Natural Sciences and Mathematics, Ss. Cyril and Methodius University, Gazi Baba bb, 1000 Skopje, North Macedonia; 10Faculty of Sciences and Mathematics, University of Niš, Višegradska 33, 18000 Niš, Serbia; 11School of Forestry and Natural Environment, Aristotle University of Thessaloniki, 54124 Thessaloniki, Greece; 12Faculty of Agriculture, University of Zagreb, Svetošimunska 25, 10000 Zagreb, Croatia

**Keywords:** AFLP, Balkan Peninsula, BAPS, essential oils, Mantel test, PCA, STRUCTURE

## Abstract

A study of the phytochemical and molecular characteristics of ten *Micromeria* and six *Clinopodium* taxa (family Lamiaceae) distributed in the Balkan Peninsula was carried out. The phytochemicals detected in essential oils by gas chromatography, mass spectrometry, and molecular data amplified fragment length polymorphism were used to study the taxonomic relationships among the taxa and the correlations between phytochemical and molecular data. STRUCTURE analysis revealed three genetic groups, while Bayesian Analysis of Population Structure grouped the studied taxa into 11 clusters nested in the groups obtained by STRUCTURE. Principal components analysis performed with the 21 most represented compounds in the essential oils yielded results that were partly consistent with those obtained by STRUCTURE and neighbour-joining analyses. However, their geographic distributions did not support the genetic grouping of the studied taxa and populations. The Mantel test showed a significant correlation between the phytochemical and genetic data (r = 0.421, *p* < 0.001). Genetic distance explained 17.8% of the phytochemical distance between populations. The current taxonomic position of several of the studied taxa is yet to be satisfactorily resolved, and further studies are needed. Such future research should include nuclear and plastid DNA sequences from a larger sample of populations and individuals.

## 1. Introduction

The genus *Micromeria* Benth. (Lamiaceae) includes 54 [1], 70, or only 20 [2] annual and perennial herbs, sub-shrubs, and shrubs, depending on the point of view. According to Bräuchler et al. [1,3], the distribution of *Micromeria* species extends from the Mediterranean to South Africa and Madagascar and from China to the Macaronesian Archipelago. Chater and Guinea [4] described 21 *Micromeria* species for Europe, with more than half of these species occurring in the Balkan Peninsula. The genus *Micromeria* is part of a complex group of genera in the tribe *Mentheae* and subtribe *Menthinae* (subfamily Nepetoideae, family Lamiaceae) and has often been considered part of the loosely defined “Satureja” complex [5,6,7]. On the other hand, Bentham [8] accepted *Micromeria* species strictly as a distinct genus, and this opinion prevails today [9,10,11,12]. Harley et al. [13] proposed four sections (Micromeria, Pineolentia, Cymularia, and Pseudomelissa) for the infrageneric subdivision of the genus *Micromeria*, while Doroszenko [10] described the morphological traits of those sections. The species of genus *Micromeria* inhabiting the Balkan Peninsula, which are the focus of this article, belong to the sections Pseudomelissa and Micromeria [13] or Pseudomelissa and Eumicromeria [14]. Bräuchler et al. [3] showed that the genus *Micromeria* is polyphyletic and that its revision is necessary. Based on this view, Bräuchler et al. [15] transferred the section Pseudomelissa from the genus *Micromeria* to the genus *Clinopodium* L. Bräuchler et al. [1] also provided a comprehensive list of new combinations, synonyms, and valid names. Studies on micromorphological characters of Balkan *Micromeria* and closely related *Clinopodium* species support the recent transfer of the section Pseudomelissa into the genus *Clinopodium* [16].

The genus *Micromeria* is represented in the Balkan Peninsula by several species with narrow distribution ranges that are primarily included on local floras and lists [17,18,19,20,21,22,23]. Their taxonomic position, based on morphological characters, has not always been clearly defined, with a variable, complex synonymy, and author subjectivism. Despite a new taxonomic position of the section Pseudomelissa, many authors [24,25,26,27,28,29,30] still use the previously assigned names.

Genetic studies can help reconstruct the evolutionary history and the delimitation of species or subspecies [31,32]. The evolutionary history and underlying genetic structure of closely related taxa may be confronted with different habitats that impose particular environmental constraints upon them [33]. Although genetic analyses provide the most helpful information for taxonomic studies today, chemical investigations such as the detection of essential oils (EO) or phenolic substances can also help resolve taxonomic uncertainties [34,35,36]. The use of phytochemicals as taxonomic characters in resolving issues in plant taxonomy has been addressed by several authors [37,38,39,40,41]. The EO content in *Micromeria* and *Clinopodium* species have been widely studied [24,25,27,30,42,43,44]. However, only a few studies [34,42] have aimed to find connections between EO content and the species’ taxonomic positions.

This study aims to obtain additional knowledge about *Micromeria* and closely related *Clinopodium* species recently transferred from the section Pseudomelissa that are widely distributed in the Balkan Peninsula. To achieve this goal, molecular and phytochemical studies were performed at the population level on 16 Balkan *Micromeria* and closely related *Clinopodium* taxa.

## 2. Results

### 2.1. AFLP Analysis

Amplified fragment length polymorphism (AFLP) analysis revealed high contrasts in population–genetic parameters among the studied populations. The percentage of polymorphic fragments and Shannon’s index (Table 1) varied among the studied taxa, with the lowest values observed in *Clinopodium pulegium* (Cp) (13.75%; 0.126) and the highest in *Micromeria cristata* ssp. *cristata* (population Mc3) (41.68%; 0.287). Of the 1694 polymorphic markers in 434 individuals, 19 were private (unique to a specific population). They were detected across 14 populations, most belonging to *M. cristata* ssp. *cristata* (six private alleles) and a single population of *M. graeca* ssp. *fruticulosa* (four private alleles), whereas no private alleles were detected in populations of *M. kerneri*. Of the 16 *Clinopodium* populations, private alleles were detected in only three. Frequency down-weighted marker values (DW) ranged from 1139.31 (*M. croatica*, population Mcr2) to 6622.21 (*M. graeca* ssp*. fruticulosa*). The highest values were detected in *M. graeca* and *M. cristata*, while much lower values were found in the other studied taxa. The expected heterozygosity (*H*_E_) levels ranged from 0.073 (populations Mcr1 and Mcr5 of *M. croatica*) to 0.130 (*M*. *cristata* ssp. *kosaninii*, McrK) and 0.131 (population Mj6 of *M. juliana*). With an average value of 0.114, *M. cristata* and *M. juliana* had the highest *H*_E_ levels, while *M. longipedunculata* and *M. croatica* were on the other side of the spectrum, with an average *H*_E_ value of 0.084. Of the total variability, 48.68% refers to variability between and 51.32% within populations, indicating significant differences between the studied populations. AMOVA analysis showed that intrapopulation variability was considerably higher than among populations (Table 2). Variability among populations ranged from 5.04% (*M. graeca* ssp. *graeca*) to 34.45% (*M. kerneri*). For most species, interpopulation variability was approximately 10%. The exceptions were the populations of *M. cristata* (24.82%) and *M. kerneri* (34.45%), which had a higher and significant interpopulation distance (ΦST).

The results of neighbour-joining (NJ) analysis are shown in Figure 1. Three genetic groups determined by STRUCTURE were marked using the same colours (blue, green, red) in the NJ tree to allow comparison of the two analyses. The NJ analysis gives significant bootstrap support to confirm that *Micromeria* and *Clinopodium* are well-differentiated groups of closely related taxa. In addition, most individuals in both groups were well-supported, except for the *M. juliana*–*M. kerneri*–*M. microphylla* cluster, characterised by low differentiation among the adopted taxa. Individuals of *M. juliana* were grouped in the same cluster with *M. kerneri* from the Croatian population Gradina (Mk3). The population of *M. microphylla* (Mm) was divided into two clusters and associated with four populations of *M. kerneri*. The AMOVA results showed no statistically significant differences between *M. microphylla* and *M. kerneri*. Two populations of *M. croatica*, known from local Balkan literature as *M. pseudocroatica* (McrP), were separated from the remaining ten populations of *M. croatica* (Mcr) (Figure 1, Table 3). Additionally, the population of *M. graeca* ssp. *fruticulosa* (MgF) was recognised as a distinct taxon, separated from the two populations of *M. graeca* ssp. *graeca* (Mg). In the NJ tree, the population of *M. cristata* ssp. *kosaninii* (McK) and two populations of *M. cristata* ssp. *cristata* (Mc3, Mc5) were also separated.

The position of the studied taxa within the genus *Clinopodium* is complex. *Clinopodium dalmaticum* showed genetic differentiation into two subgroups. The first subgroup was formed by two Montenegrin populations (Cd), while the second was formed by three Bulgarian populations (CdB). Moreover, the Bulgarian populations were closer to *C*. *frivaldszkyanum* (Cf) than to the Montenegrin populations of the same species (Figure 1). Three other *Clinopodium* species studied (*C. serpyllifolium*, *C. pulegium*, and *C. thymifolium*) were separated from the other taxa with high bootstrap support.

In the STRUCTURE analysis, the highest K value was observed for K = 3 (K = 1394.43), indicating the presence of three genetic clusters (Appendix A). STRUCTURE analysis (Figure 2A) revealed three genetic groups, shown in blue, green, and red on the NJ tree (Figure 1). The first group included populations of *Micromeria cristata*, *M. croatica*, *M. graeca*, and *M. longipedunculata*; the second included the populations of *M. juliana* and *M. kerneri*; the third included all *Clinopodium* populations studied. While low levels of admixture characterised most of the studied populations, this was not the case with the *M. microphylla* population that was positioned between the two *Micromeria* clusters with high admixture levels. The genetic clusters were not geographically defined, since representatives of each cluster are found throughout the studied area.

On the other hand, BAPS analysis (Figure 2B) revealed a congruent assignment of the studied *Micromeria* and *Clinopodium* taxa to 11 clusters nested within the groups identified by the STRUCTURE analysis. The best partitions received log-likelihoods of −182,699.06 at P = 1 (without using geographic coordinates as informative priors) and −183,284.74 at P = 1 (with spatial clustering). In general, both methods produced nearly identical results. The first two groups of the BAPS analysis were formed by *M. cristata* ssp. *cristata* (Mc) and *M. cristata* ssp. *kosaninii* (McK); the third group was formed by *M. croatica* (Mcr); the fourth by two populations of *M. croatica* covered under the disputed name *M*. *pseudocroatica* (McrP); the fifth and sixth clusters were formed by *M. graeca* ssp. *graeca* (Mg) and *M. graeca* ssp*. fruticulosa* (MgF), respectively; the seventh by *M. juliana* (Mj) and one population of *M. kerneri* (Mk3); the eighth by *M. microphylla* (Mm) and four populations of *M. kerneri* (Mk1–Mk3, Mk5); the ninth by *M. longipedunculata* (Ml); the tenth by *C. pulegium* (Cp), *C. serpyllifolium* (Cs), *C. dalmaticum* (Cd), and *C. thymifolium* populations; the eleventh cluster was formed by *C. frivaldszkyanum* (Cf) and three *C. dalmaticum* (CdB) populations listed under the contested name *M. bulgarica*.

### 2.2. Essential Oil Analysis

The composition and yield of EO in this study included 41 oil samples from the genus *Micromeria* and 15 samples from the genus *Clinopodium* (Appendix A). The yields of the studied taxa ranged from a minimum of 0.35% in *C. frivaldszkyanum* to 1.79% in *M. croatica.* The composition of EO of all studied taxa can be divided into the following classes: monoterpene hydrocarbons (1.66–44.73%), oxygenated monoterpenes (10.42–85.31%), sesquiterpene hydrocarbons (1.09–35.73%), oxygenated sesquiterpenes (0.25–39.39%), phenolic compounds (0–25.73%), carbonyl compounds (0–2.56%), and hydrocarbons (0.25–14.07%).

Further presentation of the results focuses on the main volatile components in the composition of the EO of the studied species. In the EO composition of the five populations of *Micromeria cristata* ssp. *cristata*, the compounds borneol (14.11–26.28%) and α-cadinol (12.48–17.72%) were most abundant. The compounds α-muurolol (17.53%) and pulegone (8.91%) were detected only in *M. cristata* subsp. *kosaninii*. The detection of verbenone, camphene, bornyl acetate, and α-humulene in all populations of *M. cristata* but not in *M. cristata* subsp. *kosaninii* (Appendix A) was also notable. The compounds borneol (16.13–28.71%), *E*-caryophyllene (7.13–16.8%), caryophyllene oxide (10.92–15.75%), and germacrene D (2.95–14.12%) were the main components of EO extracted from all 12 samples of *M. croatica* (Appendix A). In the EO of *M. graeca* ssp. *graeca*, α-bisabolol was present in a high percentage (23.02% and 25.78%) at both sites. In the EO of *M. graeca* subsp. *fruticulosa*, α-bisabolol was also abundant (11.92%), but the most representative compound was pinocarvone (17.39%) (Appendix A). The most abundant compounds in the ten EO samples of *M. juliana* were *E-*caryophyllene (10.62–22.35%) and caryophyllene oxide (22.26–32.72%) (Appendix A). Caryophyllene oxide (12.81–23.46%) was also the most abundant compound in the five EO samples of *M. kerneri*, followed by α-pinene (12.3–16.13%) (Appendix A). Isolates of the species *M. longipedunculata* contained the most spathulenol (more than 30% of all four populations studied), while *M. microphylla* was rich in eudesem-7(11)-en-4-*ol* (22.91%) (Appendix A).

*Clinopodium dalmaticum* showed the most significant differences in EO composition among the investigated *Clinopodium* taxa. The oil composition in the Montenegrin samples consisted mainly of piperitone (more than 30%). Two Bulgarian populations presented high concentrations of *E*-caryophyllene (CdB1, 31.74%; CdB2, 42.43%), while population CdB3 had the highest content of *α*-pinene (14.31%). Thymol was also prominent in Bulgarian populations of *C. dalmaticum*. Pulegone (Cf1, 47.2%; Cf2, 29.52%) and menthone (Cf1, 12.8%; Cf2, 9.23%) were dominant in the composition of *C. frivaldszkyanum* (Appendix A). In all other *Clinopodium* species studied (*C. pulegium*, *C. serpyllifolium*, and *C. thymifolium*) the most abundant compounds were pulegone and piperitenone oxide, making these oils extremely rich in oxygenated monoterpenes (64.99–85.31%) (Appendix A).

PCA analysis (Figure 3) was performed on the 21 compounds isolated from the EO, in amounts exceeding 10% per sample (population). PC1 and PC2 for the EO compounds explained 29.26% of the variance. Pearson’s correlation coefficients between 21 main compounds and scores of the first two PC are shown in Table 4. The phytochemical groups obtained by the PCA were partly similar to the three genetic groups determined by NJ and STRUCTURE analyses. Among the 21 compounds, PCA detected nine components that contributed most to the differences between groups (Figure 3). *Clinopodium* species were mainly located in the negative region of PC1 and PC2, while *Micromeria longipedunculata* (Ml) was positioned among the *Clinopodium* species.

The main compounds in this group were menthone, pulegone, and piperitenone oxide, and the highest values were found in *Clinopodium frivaldszkyanum* (Cf1, menthone) and *C. thymifolium* (Ct1, pulegone; Ct7, piperitenone oxide) (Appendix A). Only *C. dalmaticum* from Bulgaria (formerly *Micromeria bulgarica*, CdB) had an unusual position among the *Clinopodium* species, positioned near *M. kerneri* and *M. juliana* in the negative region of PC1 and the positive region of PC2. The specific compounds for this phytochemical group were verbenone, caryophyllene oxide, and docosane, which were highest in three populations of *M. juliana* (Mj7, Mj8, Mj5) (Appendix A).

Populations belonging to the third phytochemical group were primarily positioned in the positive region of PC1 and PC2 (Figure 3). Camphor, borneol, and germacrene D were the components distinguishing this group, and were highest in *Micromeria croatica* (Mcr10, germacrene D) and the contested taxon *M*. *pseudocroatica* (McrP1, camphor; McrP2, borneol) (Appendix A).

### 2.3. Mantel Test

The correlations between AFLP and the phytochemical matrices of dissimilarity were calculated using the Mantel test. A significant correlation was observed between the phytochemical and molecular data (r = 0.421, p_Mantel_ < 0.001). According to the same test, 17.8% (R = 0.178) of the phytochemical distance between populations could be explained by genetic distance (Appendix A). 

## 3. Discussion

Several phylogenetic conclusions can be drawn from the genetic diversity and STRUCTURE results. The NJ analysis separated *Micromeria* from *Clinopodium* taxa and reinforced the recent transfer of species from the section Pseudomelissa (genus *Micromeria*) to the genus *Clinopodium* by Bräuchler et al. [15]. Although the distinction was confirmed between the *Micromeria* and *Clinopodium* groups, it is questionable whether it is substantial enough to label these groups as separate genera. STRUCTURE analysis indicated the presence of three genetic clusters: two within *Micromeria* and a third of *Clinopodium* species. If *Clinopodium* is treated as a separate genus, then the other two *Micromeria* groups should also be considered as such. Since the previous *Clinopodium–Micromeria* segregation was based on a small sample size and analysis of a single cpDNA region [3], the results presented here are even more relevant.

The NJ analysis showed genetic differentiation of *Clinopodium dalmaticum* in the Montenegrin and Bulgarian populations. Regarding the EO composition, PCA also separated Montenegrin from Bulgarian populations of *C. dalmaticum* (Figure 3). These results suggest variability within *C. dalmaticum*, with the note that the Bulgarian populations were previously considered to be *Micromeria bulgarica* [19,20]. Variability within *C. dalmaticum* was previously described by Vandas [45], who identified *M. bulgarica* in the area of Usunža and Krivska River (North Macedonia). Chater and Guinea [4] and Ančev [19] identified two subspecies of *M. dalmatica* (now *C. dalmaticum*): *M. dalmatica* ssp. *dalmatica* and *M. dalmatica* ssp. *bulgarica* (Velen.) Guinea. On the other hand, Bräuchler et al. [1,15] concluded that *M. bulgarica* is a synonym of *C. dalmaticum*. Since *C. dalmaticum* showed both molecular and phytochemical separation among its populations, research on this species should be continued. The Bulgarian population of *C. dalmaticum* was closer to *C. frivaldszkyanum* than to the Montenegrin populations of *C. dalmaticum* (Figure 1). This suggests that the taxonomic relationships within these two or three taxa of the genus *Clinopodium* require further clarification. Given the considerable geographic distance between the Montenegrin and Bulgarian populations of *C. dalmaticum*, the existence of two geographically distinct groups is not unusual. Such a refugia-within-refugia model developed by Gómes and Lunt [46] for the Iberian Peninsula has also been applied to some species from the Balkan Peninsula [47,48,49].

In studies of individual species or groups of closely related species in the Balkan Peninsula, the question of the presence of (micro)refugial areas that protected local populations during unfavourable climatic conditions of glaciation cycles cannot be avoided. In the AFLP analyses, DW markers (Table 1) are considered indicators that detect such areas. The results presented here are partially contradictory, making it difficult to draw a plausible conclusion. The highest values of DW markers were observed in *Micromeria graeca* populations from the central Adriatic islands, followed by several *M. cristata* populations from the eastern and central Balkans (Mc2, Mc3, and McK) and a single population of *M. croatica* (McrP2) from the southern Adriatic mainland. Populations characterised by moderately high levels of DW markers were more common than populations with highest values. They were also scattered over a large area, ranging from the eastern Adriatic coastal region to the eastern Balkans.

Similar to the STRUCTURE results, no spatial structuring was observed of populations characterised by high frequencies of DW markers, suggesting that no single refugial area can be identified within the Balkan Peninsula. Instead, there appear to have been numerous microrefugia scattered over large areas. The northern Adriatic coastal area does not harbour any of this microrefugia, as low levels of these markers characterised these populations.

The NJ analysis separated *Micromeria* species into several genetic groups and raised questions about the systematic position of certain taxa. Populations of *M. cristata* were separated into two statistically significant different subgroups (Figure 1, Table 3). One subgroup encompassed two western populations of *M. cristata* ssp. *cristata* (Mc3, Mc5) and *M. cristata* ssp. *kosaninii* (McK), while the second subgroup was formed by three eastern populations of *M. cristata* ssp. *cristata* (Mc1, Mc2, Mc4). On the other hand, the PCA analysis based on phytochemical compounds separated *M. cristata* ssp. *kosaninii* from populations of *M. cristata* ssp. *cristata* (Figure 3). A possible reason for this is in the different habitat conditions, as these can affect EO content [50].

Two populations of *Micromeria croatica* (McrP1, McrP2), known in the Balkan literature [18,23] under the name *M. pseudocroatica* (McrP), were separated from the remaining ten populations of *M. croatica* (Mcr1–Mcr10) with high bootstrap support. The PCA analysis of phytochemical traits also split *M. croatica* (Mcr) from the disputable taxon *M. pseudocroatica* (McrP) (Figure 3). Additionally, the difference between *M. croatica* and *M. pseudocroatica* was greater than the difference among the ten populations of *M. croatica*. Although Bräuchler et al. [1] concluded that *M. pseudocroatica* is only a synonym of *M. croatica*, future genetic research should verify whether the differences presented herein might be due to geographical isolation. Namely, populations of the disputable *M. pseudocroatica* are located on the Pelješac Peninsula and are partly isolated from continental populations of *M. croatica*.

The population of *Micromeria microphylla* (Mm) was divided into two clusters associated with four populations of *M. kerneri*, although the differences between the species were not statistically significant (Table 3). On the other hand, the PCA analysis of EO compounds showed that *M. microphylla* (Mm) is quite different from populations of *M. kerneri* (Figure 3). Different habitat conditions can also explain the differences in EO composition between these species. The complexity in this group is further increased by one population of *M. kerneri* (Mk3), which is divided into two clusters closely related to *M. juliana* (Figure 1). The AMOVA (Table 3) showed a difference between *M. juliana* and *M. kerneri* regardless of the presence of the disputable population Mk3. The PCA analysis of phytochemicals also suggests difference between *M. kerneri* and *M. juliana* (Figure 3). On the other hand, Bräuchler et al. [1] considered that *M. juliana* and *M. kerneri* might be conspecific. The obtained results indicate molecular and phytochemical differences between *M. kerneri* and *M. juliana*. The additional molecular analysis should check whether the Balkan populations of *M. kerneri* belong to *M. juliana* and examine their relationship to *M. microphylla*.

STRUCTURE analysis (Figure 2A) detected a general distribution in three genetic groups. Their geographic distributions did not support the genetic grouping of the studied taxa and populations. Surprisingly, there was virtually no spatial structuring of the recognised genetic clusters from the STRUCTURE analysis, as representatives from all groups are found across large regions, mixed in a seemingly random fashion. Such a result is hard to explain, but it confirms that the distribution ranges of the studied taxa do not follow the levels of their relatedness. It should be noted that a similar situation was also detected, where numerous populations of genetically well-supported taxa were scattered over large areas without any signs of spatial groupings. Such a result suggests the presence of strong gene flow barriers among closely related taxa, eliminating any possibility for interspecies hybridisation and consequent fusion of these taxa into spatially structured clusters. The exception is mentioned in the *Micromeria juliana–M. kerneri–M. microphylla* complex, where these barriers are weak at best. As such, there is currently no clear explanation for the obtained results. This is possibly a consequence of contrasting evolutionary histories and environmental conditions experienced by these taxa that have resulted in the development of strong reproductive isolation mechanisms.

Another set of results enabled a more straightforward conclusion. Not only were the majority of analysed taxa well-supported (except for the *Micromeria juliana–M. kerneri–M. microphylla* complex), but the recognition of a few additional taxa is now possible, thus opening the possibility for systematic repositioning within the studied groups. Within the *C. dalmaticum* group, two well-supported taxa were identified, one formerly known as *M. dalmatica* and the second as *M. bulgarica*. Within the *Micromeria* group, a similar result was observed in three cases. *Micromeria graeca* ssp*. fruticulosa*, formerly known as *M. fruticulosa,* emerged as a well-supported taxon and not as a representative of *M. graeca.* Similarly, two populations of *M. croatica*, formerly recognised as *M. pseudocroatica,* were differentiated from any other taxon, thus confirming their status as a separate species. Within the *M. cristata* group, two taxa have emerged: one comprising populations Mc3, Mc5, and McK, and another comprising Mc1, Mc2, and Mc4. *Micromeria cristata* ssp. *kosaninii* (McK), formerly recognised as *M. kosaninii,* seems to lack the support needed for its recognition as either a species or subspecies. However, bearing in mind that Bräuchler et al. [1] validated a new combination of *M. cristata* ssp. *kosaninii*, future research should also examine its taxonomic position and whether certain western populations of *M. cristata* ssp. *cristata* belong to a new combination, *M. cristata* ssp. *kosaninii*.

EO content is known to depend on the developmental stage of the plant and the collection site [50]. To exclude the influence of the plant’s developmental stage, the aboveground plant parts of all investigated taxa were collected for isolation of EO during flowering time. The composition of EO of *Micromeria* and *Clinopodium* taxa were investigated in all populations of the studied taxa. In general, the results presented in this study are consistent with patterns reported in the literature. In the composition of EO of *M. cristata* collected in Serbia, Bulgaria, Greece, and North Macedonia, the most abundant compound was borneol (14.11–26.28%) (Appendix A). Kostadinova et al. [51] also identified borneol (6.1%) in a sample of *M. cristata* from Bulgaria, while its isomer isoborneol (11.3%) was the most abundant in the sample collected in Serbia [52]. The extent to which subspecies can differ in EO composition was shown by Çarikçi [53], who studied three subspecies of *M. cristata*, namely *M. cristata* ssp. *cristata, M. cristata* ssp. *phyrigia* P. H. Davis, and *M. cristata* ssp. *orientalis* P. H. Davis. In all three subspecies, the main constituents of EO were borneol and camphor [53]. Thus, the compounds α-muurolol and pulegone that were predominant in the taxon *M. cristata* ssp*. kosaninii* (Appendix A) were not identified in the subspecies from Turkey. These differences are not surprising, considering the geographical distance and habitat conditions.

Borneol was also one of the main compounds in the studied samples of *Micromeria croatica* (Appendix A) from Croatia, Bosnia and Herzegovina, and Montenegro, followed by the compounds *E*-caryophyllene and caryophyllene oxide. Caryophyllene oxide was the main compound in most of the studied populations of *M. croatica*, according to Slavkovska et al. [34], Kremer et al. [54], and Vuko et al. [55]. The EO of *M. graeca* ssp. *graeca* analysed here was rich in *α*-bisabolol (Appendix A), while in the same taxon from Greece, the main component was *epi-α*-bisabolol [56]. In the composition of the ten EO samples of *M. juliana*, the most abundant volatile components were *E-*caryophyllene and caryophyllene oxide (Appendix A). Similarly, these two compounds also dominated the EO composition of *M. juliana* from Anatolia, Turkey [53]. Caryophyllene oxide (12.81–23.46%) was the most abundant compound in the five *M. kerneri* oils studied, followed by *α*-pinene (12.3–16.13%) (Appendix A). A previous study also showed that the EO composition of *M. kerneri* and *M. juliana* was characterised by a high concentration of oxygenated sesquiterpenes, with caryophyllene oxide as the most abundant compound [42]. In this study, the composition of EO of *M. microphylla* was reported for the first time, dominated by eudesem-7-(11)-en-4-*ol* (22.91%) (Appendix A). The peculiarity of the oil composition is not surprising, considering the isolation of this population in the central Adriatic (Table 1, Figure 4).

*Clinopodium dalmaticum* is endemic to the Balkan Peninsula, and is widespread in Bulgaria, Montenegro, and Greece, including Crete [57]. In this study, the volatile compounds of samples collected in Montenegro and Bulgaria were identified. In the composition of the isolates from Montenegro, the predominant compound was piperitone, while the Bulgarian samples were rich in *E*-caryophyllene, *α*-pinene, and thymol (Appendix A). The most abundant compounds in the EO of *C. frivaldszkyanum, C. pulegium*, *C. serpyllifolium*, and *C. thymifolium* were pulegone and piperitenone oxide, making oils of these species extremely rich in oxygenated monoterpenes (53–85.31%) (Appendix A). According to Zheljazkov [58], pulegone was one of the main constituents in the EO of *C. frivaldszkyanum* from the Bulgarian populations of Shipka and Uzana.

The Mantel test showed a significant correlation between the phytochemical and AFLP data (r = 0.421, *p* < 0.001). The literature on this topic is diverse. According to Slavkovska et al. [34], the composition and quantity of EO of *Micromeria* species distinguished section Pseudomelissa from Eumicromeria. The EO of species from the section Pseudomelissa was dominated by oxygenated monoterpenes of the menthane type, while various terpene compounds were dominant in species from the section Eumicromeria [34]. Multivariate analysis (PCA and UPGMA) of compositions determined in the EO of *M. kerneri* and *M. juliana* separated the populations of these two species [42]. Feulner et al. [36] determined strong and significant correlations between AFLP data and floral scent volatiles at the population level (r = 0.791, *p* = 0.004) and individual level (r = 0.823, *p* < 0.001) in *Sorbus* taxa (family Rosaceae). Xavier et al. [59] found a significant correlation between volatile chemical classes and genetic traits of *Aniba* Aubl. species in the Amazon region in Pará State (Brazil). Additionally, AFLP profiles of 11 *Hypericum* species and cultivars were correlated with their levels of phytochemical markers (chlorogenic acid, hyperforin, hypericin, pseudohypericin, and rutin) determined in their methanolic extracts enabling true-to-type identification and marker-assisted breeding programmes [60]. Investigations of 20 populations of four *Thymus* L. species native to Hungary found only partial similarities between dendrograms generated by hierarchical cluster analysis based on DNA patterns and EO samples [61]. On the other hand, in a study of *Ophrys* L. taxa (Orchidaceae), Stökl et al. [62] did not find any correlation between scent data and DNA-molecular data. Trindade et al. [63,64] concluded that there was no correlation between the chemical analysis of EO of *Thymus caespititius* Brot. from the Azores and molecular data sets. Volatile and molecular analysis of *Juniperus brevifolia* (Seub.) Antoine from the same archipelago also showed no correlation between chemical and molecular data sets [65]. Finally, Emami-Tabatabaei et al. [66] studied the possible correlation between AFLP data and the EO profile obtained by GC-MS of *Lutea elbursensis* Mozaff from northern Iran, concluding that the chemical composition of EO cannot be used as a reliable taxonomic tool.

## 4. Materials and Methods

### 4.1. Plant Material

Randomly selected samples of wild-growing plants of *Micromeria* and closely related *Clinopodium* species were collected during the blooming period from June to August 2018. Voucher specimens of herbal material were deposited in the Fran Kušan Herbarium, Faculty of Pharmacy and Biochemistry, University of Zagreb, Croatia (Table 1, Figure 4). For molecular analysis, several young leaves from 3 to 11 plants per population were collected on a dry day. Immediately after collection, leaves were dried in plastic bags containing silica gel and stored for further use in DNA analysis. Additionally, above-ground shoots with leaves and flowers were harvested and mixed to obtain a randomly selected sample. The collected plant parts were air-dried and protected from direct sunlight for 15 days at 22 °C and 60% relative humidity. From each locality, 50 g of air-dried plant material was hydro-distilled for 3 h in a Clevenger-type apparatus. The EO obtained was dried over anhydrous sodium sulphate.

### 4.2. Molecular Analysis

#### 4.2.1. DNA Isolation

Genomic DNA was isolated using a commercial DNA isolation kit (GenElute™ Plant Genomic DNA Miniprep Kit, Sigma-Aldrich^®^, Darmstadt, Germany), while a nanophotometer P330 (Implen^®^, München, Germany) was used to measure DNA concentrations and quality. The AFLP technique [67] was carried out according to the modified protocol described by Carović-Stanko et al. [68]. Four primer combinations were used for selective amplification: VIC-*Eco*RI-ACG + *Tru*1I-CGA, NED-*Eco*RI-AGA + *Tru*1I-CGA, FAM-*Eco*RI-ACA + *Tru*1I-CGA, and PET-*Eco*RI-ACC + *Tru*1I-CGA.

#### 4.2.2. AFLP Data Analysis

##### Within-Population Diversity

To construct a binary matrix, the obtained AFLP fragments were scored as present (1) or absent (0). Diversity within populations was assessed by calculating the proportion of polymorphic markers (%*P*), the number of private markers (*N_pr_*), and the frequency down-weighted marker values (DW) [69] using the AFLPdat package in R [70]. The Shannon information index of each population was calculated as *I* = −Σ (*p_i_* log_2_ *p_i_*), where *p_i_* is the phenotypic frequency [71,72]. In addition, genetic diversity (*H_E_*) was calculated using a Bayesian approach [73], assuming the Hardy–Weinberg equilibrium due to outcrossing (*F_IS_* = 0) as implemented in AFLP-Surv v. 1.085 (Vekemans, X., Laboratoire de Génétique et Ecologie Végétale, Université Libre de Bruxelles, Bruxelles, Belgium) [74]. The overall mismatch error rate for all AFLP primer combinations was 2.5%.

##### Population Differentiation and Structure

Using the pairwise distance matrix based on the Dice coefficient [75], a neighbour joining tree was constructed and bootstrapped using 1000 replicates [76] using PAST v2.01 (Hammer, Ø., Paleontological Museum, University of Oslo, Oslo, Norway) [77].

Analysis of molecular variance (AMOVA) [78] was used to partition the total genetic variance among and within populations of each taxon and between closely related taxa, among populations within taxa and within populations. The variance components were tested with 10,000 permutations in Arlequin ver. 3.5.2.2 (Excoffier, L., Lischer, H., Institute of Ecology and Evolution, University of Berne, Bern, Switzerland) [79].

Population structure was assessed using two Bayesian clustering approaches implemented in STRUCTURE v2.3.4 (Pritchard Lab., Stanford University, Stanford, CA, USA) [80] and BAPS v6.0 (Corander, J., Cheng, L., Marttinen, P.; Sirén, J.; Tang, J., Department of Mathematics and Statistics, University of Helsinki, Helsinki, Finland) [81]. In STRUCTURE, 30 runs were performed for each K by setting the number of clusters (K) from 1 to 21. Each run consisted of a burn-in period of 200,000 steps followed by 1,000,000 Monte Carlo Markov Chain (MCMC) replicates assuming an admixture model and correlated allele frequencies. The calculations were performed on the Isabella computer cluster at the University of Zagreb, University Computing Centre (SRCE). The most probable number of K was selected by calculating ΔK [82] in StructureSelector [83]. StructureSelector was also used to cluster and average the results of independent runs using the approach described by Kopelman et al. [84]. BAPS was applied for population mixture analysis without the geographic origin of samples as an informative prior (‘Clustering of Individuals’) and with this prior (‘Spatial Clustering of Individuals’) [85]. The maximum number of clusters (K) was set to 20, and each run was replicated 10 times. Population admixture analysis [86] was performed with the default settings.

### 4.3. Gas Chromatography and Mass Spectrometry (GC-MS) Analyses

The EO of each *Micromeria* and *Clinopodium* sample obtained by hydro-distillation were collected for each sample in a pentane/diethyl ether mixture and analysed by gas chromatography and mass spectrometry (GC-MS). GC was performed using a gas chromatograph (model 3900; Varian Inc., Lake Forest, CA, USA) and a mass spectrometer (model 2100T; Varian Inc.). The MS conditions were ion source temperature 200 °C, ionisation voltage 70 eV; mass scan range: 40–350 mass units. The carrier gas was helium. Two columns were used: nonpolar VF-5 ms and polar capillary columns CP Wax 52. The conditions for the VF-5 ms column were temperature 60 °C isothermal for 3 min, then increased to 246 °C at a rate of 3 °C min^−1^, and held isothermal for 25 min. The CP Wax 52 column conditions were: temperature 70 °C isothermal for 5 min, then increased to 240 °C at a rate of 3 °C-min^−1^, and held isothermal for 25 min. The injection volume was 2 μL and the split ratio was 1:20. The triplicate analyses of individual peaks were identified by comparing the retention indices of the n-alkanes with literature data and authentic standards [87,88].

### 4.4. Principal Component Analysis

Principal component analysis (PCA) was based on 21 significant constituents of the EO. PCA was performed using the PRINCOMP procedure in SAS v9.3 (SAS Institute Inc., Cary, NC, USA) [89], and the biplot showing the populations and oil constituents (as vectors) was constructed using the first two principal components.

### 4.5. Mantel Test

The Mantel test [90] was used to test the correlation between genetic and biochemical data matrices. Pairwise genetic distances between populations were calculated using Nei’s standard genetic distance (DNei) in AFLP-Surv v1.085 [69]. Biochemical differences were calculated as Euclidean distances between populations for the first two principal components of the PCA of EO constituents. The significance level was assessed after 10,000 permutations in NTSYS-pc v2.21L [91].

## 5. Conclusions

STRUCTURE analysis based on AFLP genetic data grouped the studied ten *Micromeria* and six closely related *Clinopodium* taxa distributed in the Balkan Peninsula into three genetic groups. The first cluster included all *Clinopodium* taxa, while *Micromeria* species were divided into two clusters. In general, their geographic distributions did not support the genetic grouping of the studied taxa and populations. Numerous populations of genetically well-supported taxa were also found scattered over large areas with no evidence of spatial groupings. Such a result suggests that substantial gene flow barriers exist between closely related taxa, precluding any possibility for inter-species hybridisation and consequent fusion of these taxa into spatially structured clusters. An exception is the *M. Juliana–M. kerneri–M. microphylla* complex, where these barriers are weakest. Generally, groups of taxa were much less supported than individual taxa, indicating their concurrent dispersal and approximately the same time of origin.

The results also showed that the current taxonomic position of certain species requires stronger resolution. Within the *C. dalmaticum* group, two well-supported taxa were identified, one formerly known as *M. dalmatica* and the second as *M. bulgarica*. The species *M. graeca* ssp*. fruticulosa* (formerly *M. fruticulosa*) emerged as a well-supported taxon and not a representative of *M. graeca.* Two populations of *M. croatica*, formerly recognised as *M. pseudocroatica*, were also clearly differentiated from any other taxon. Within the *M. cristata* group, the taxon *M. cristata* ssp. *kosaninii* (formerly *M. kosaninii*) lacks the needed support for its recognition as either a species or subspecies. Although further studies are needed on some species within the genera *Clinopodium* and *Micromeria*, the AFLP data obtained in this research provide a good starting point for future studies. Such a study should include nuclear and plastid DNA sequences on a larger sample of populations and individuals. Finally, the Mantel test showed a significant correlation between phytochemical and AFLP data.

## Figures and Tables

**Figure 1 plants-11-03407-f001:**
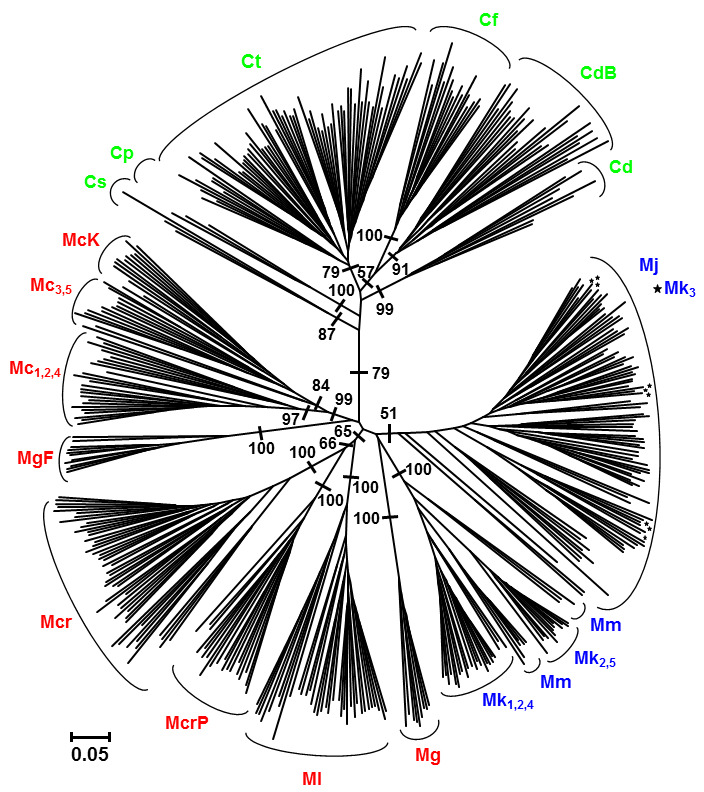
Neighbour-joining tree based on the AFLP data of *Micromeria* (*M*.) and *Clinopodium* (*C*.) taxa. Bootstrap values > 50% are indicated for major branches. Mc, *M. cristata* ssp. *cristata*; McK, *M. cristata* ssp. *kosaninii*; Mcr, *M*. *croatica* including the former *M*. *pseudocroatica*, McrP; Mg, *M. graeca* ssp. *graeca*; MgF, *M. graeca* ssp. *fruticulosa*; Mj, *M. juliana*; Mk, *M. kerneri*; Ml, *M. longipedunculata*; Mm, *M. microphylla*; Cd, *C. dalmaticum* including the former *M. bulgarica*, CdB; Cf, *C. frivaldszkyanum*; Cp, *C. pulegium*; Cs, *C. serpyllifolium*; Ct, *C. thymifolium*; three genetic groups determined by STRUCTURE were marked using the same colours (blue, green, red) in the NJ tree to allow comparison of the two analyses.

**Figure 2 plants-11-03407-f002:**
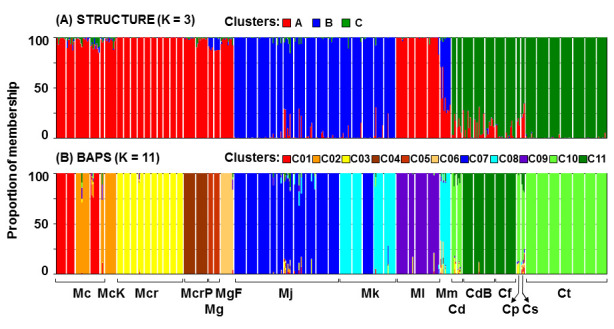
Genetic structure of the investigated taxa based on AFLP markers as resolved by Bayesian clustering. STRUCTURE assuming three clusters (**A**) while BAPS assuming 11 clusters (**B**). Mc, *M. cristata* ssp. *cristata*; McK, *M. cristata* ssp. *kosaninii*; Mcr, *M*. *croatica* including the former *M*. *pseudocroatica*, McrP; Mg, *M. graeca* ssp. *graeca*; MgF, *M. graeca* ssp. *fruticulosa*; Mj, *M. juliana*; Mk, *M. kerneri*; Ml, *M. longipedunculata*; Mm, *M. microphylla*; Cd, *C. dalmaticum* including the former *M. bulgarica*, CdB; Cf, *C. frivaldszkyanum*; Cp, *C. pulegium*; Cs, *C. serpyllifolium*; Ct, *C. thymifolium*.

**Figure 3 plants-11-03407-f003:**
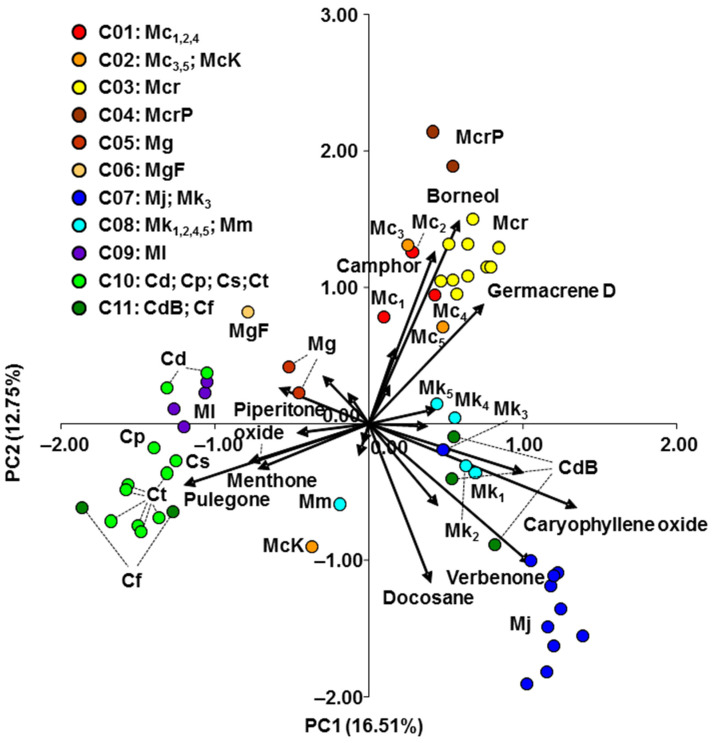
PCA of the investigated taxa based on the 21 most abundant compounds isolated from essential oil. The nine compounds contributing most to the difference between the three genetic groups are marked. Mc, *M. cristata* ssp. *cristata*; McK, *M. cristata* ssp. *kosaninii*; Mcr, *M*. *croatica* including the former *M*. *pseudocroatica*, McrP; Mg, *M. graeca* ssp. *graeca*; MgF, *M. graeca* ssp. *fruticulosa*; Mj, *M. juliana*; Mk, *M. kerneri*; Ml, *M. longipedunculata*; Mm, *M. microphylla*; Cd, *C. dalmaticum*, including the former *M. bulgarica*, CdB; Cf, *C. frivaldszkyanum*; Cp, *C. pulegium*; Cs, *C. serpyllifolium*; Ct, *C. thymifolium*.

**Figure 4 plants-11-03407-f004:**
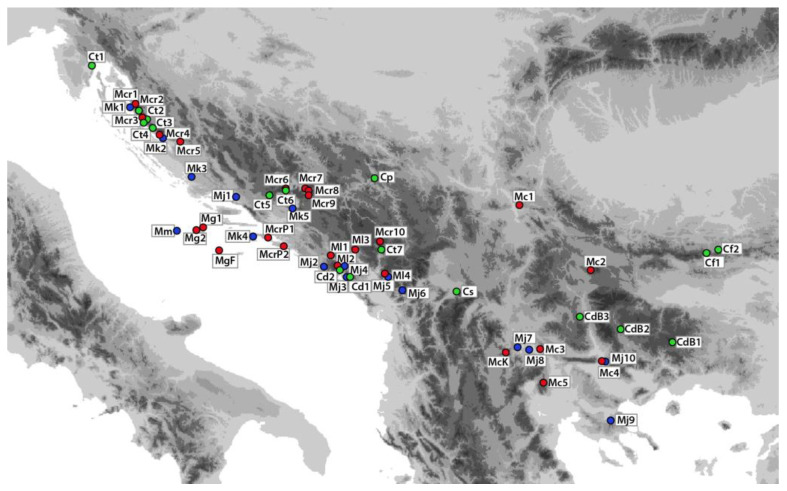
Collection sites of studied *Micromeria* and *Clinopodium* taxa: Mc, *M*. *cristata* ssp. *cristata*; McK, *M*. *cristata* ssp. *kosaninii*; Mcr, *M*. *croatica* including the former *M*. *pseudocroatica*, McrP; Mg, *M*. *graeca* ssp. *graeca*; MgF, *M*. *graeca* ssp. *fruticulosa*; Mj, *M*. *juliana*; Mk, *M*. *kerneri*; Ml, *M*. *longipedunculata*; Mm, *M*. *microphylla*; Cd, *C*. *dalmaticum* including the former *M*. *bulgarica*, CdB; Cf, *C*. *frivaldszkyanum*; Cp, *C*. *pulegium*; Cs, *C*. *serpyllifolium*; Ct, *C*. *thymifolium*. Three genetic groups determined by STRUCTURE were marked using the same colours (blue, green, red).

**Table 1 plants-11-03407-t001:** Details on origin and collection data and molecular diversity revealed by AFLP markers in *Micromeria* (*M*.) and *Clinopodium* (*C*.) taxa.

Taxa According to:			Sample Site		Voucher No.		Altitude	
[1] (Code)	Balkan Literature		(Population)				(m)	
*M. cristata* ssp. *cristata* (Mc1)	*M. cristata*		Humsko brdo Mt (Serbia)		HFK-HR-51126		387	
*M. cristata* ssp. *cristata* (Mc2)	*M. cristata*		Vitosha Mt (Bulgaria)		HFK-HR-51132		980	
*M. cristata* ssp. *cristata* (Mc3)	*M. cristata*		Demir Kapija (N. Macedonia)		HFK-HR-51141		111	
*M. cristata* ssp. *cristata* (Mc4)	*M. cristata*		Nomos Serron (Greece)		HFK-HR-51145		190	
*M. cristata* ssp. *cristata* (Mc5)	*M. cristata*		Pαikon Mt (Greece)		HFK-HR-51146		1650	
*M. cristata* ssp. *kosaninii* (McK)	*M. kosaninii*		Pletvar (N. Macedonia)		HFK-HR-51142		1020	
*M. croatica* (Mcr1)	*M. croatica*		Rossijev kuk (Croatia)		HFK-HR-51016		1641	
*M. croatica* (Mcr2)	*M. croatica*		Bačić kuk (Croatia)		HFK-HR-51012		1159	
*M. croatica* (Mcr3)	*M. croatica*		Stupačinovo (Croatia)		HFK-HR-51017		1058	
*M. croatica* (Mcr4)	*M. croatica*		Bojinac (Croatia)		HFK-HR-51013		1046	
*M. croatica* (Mcr5)	*M. croatica*		Prezid (Croatia)		HFK-HR-51010		991	
*M. croatica* (Mcr6)	*M. croatica*		Diva Grabovica (BIH)		HFK-HR-51007		251	
*M. croatica* (Mcr7)	*M. croatica*		Dubočani (BIH)		HFK-HR-51020		715	
*M. croatica* (Mcr8)	*M. croatica*		Rakitnica (BIH)		HFK-HR-51019		943	
*M. croatica* (Mcr9)	*M. croatica*		Glavatičevo (BIH)		HFK-HR-51005		366	
*M. croatica* (Mcr10)	*M. croatica*		Babji zub (Montenegro)		HFK-HR-51003		1438	
*M. croatica* (McrP1)	*M*. *pseudocroatica*		Pijavičino (Croatia)		HFK-HR-51032		443	
*M. croatica* (McrP2)	*M*. *pseudocroatica*		Prapratno (Croatia)		HFK-HR-51033		159	
*M. graeca* ssp. *graeca* (Mg1)	*M. graeca*		Malo zlo polje (Croatia)		HFK-HR-51036		137	
*M. graeca* ssp. *graeca* (Mg2)	*M. graeca*		Komiža (Croatia)		HFK-HR-51037		43	
*M. graeca* ssp*. fruticulosa* (MgF)	*M. fruticulosa*		Sušac Island (Croatia)		HFK-HR-51038		10	
*M. juliana* (Mj1)	*M. juliana*		Omiška Dinara Mt (Croatia)		HFK-HR-51048		91	
*M. juliana* (Mj2)	*M. juliana*		Sniježnica Mt (Croatia)		HFK-HR-51041		512	
*M. juliana* (Mj3)	*M. juliana*		Lastva (BIH)		HFK-HR-51049		382	
*M. juliana* (Mj4)	*M. juliana*		Lovćen Mt (Montenegro)		HFK-HR-51056		948	
*M. juliana* (Mj5)	*M. juliana*		Krivošije Mt (Montenegro)		HFK-HR-51059		194	
*M. juliana* (Mj6)	*M. juliana*		Cijevna Canyon (Montenegro)		HFK-HR-51045		157	
*M. juliana* (Mj7)	*M. juliana*		Babuna River Canyon (N. Macedonia)		HFK-HR-51052		179	
*M. juliana* (Mj8)	*M. juliana*		Rajec Reka (N. Macedonia)		HFK-HR-51168		199	
*M. juliana* (Mj9)	*M. juliana*		Cholomon Mt (Greece)		HFK-HR-51188		1100	
*M. juliana* (Mj10)	*M. juliana*		Nomos Serron (Greece)		HFK-HR-51147		182	
*M. kerneri* (Mk1)	*M. kerneri*		Zavratnica (Croatia)		HFK-HR-51018		161	
*M. kerneri* (Mk2)	*M. kerneri*		Starigrad Paklenica (Croatia)		HFK-HR-51063		83	
*M. kerneri* (Mk3)	*M. kerneri*		Gradina (Croatia)		HFK-HR-51061		207	
*M. kerneri* (Mk4)	*M. kerneri*		Korčula Island (Croatia)		HFK-HR-51064		219	
*M. kerneri* (Mk5)	*M. kerneri*		Mostar (BIH)		HFK-HR-51044		65	
*M. longipedunculata* (Ml1)	*M. parviflora*		Jazina (BIH)		HFK-HR-51046		342	
*M. longipedunculata* (Ml2)	*M. parviflora*		Krivošije Mt (Montenegro)		HFK-HR-51047		619	
*M. longipedunculata* (Ml3)	*M. parviflora*		Nikšić (Montenegro)		HFK-HR-51066		605	
*M. longipedunculata* (Ml4)	*M. parviflora*		Cijevna Canyon (Montenegro)		HFK-HR-51067		161	
*M. microphylla* (Mm)	*M. microphylla*		Svetac Island (Croatia)		HFK-HR-51039		28	
*C. dalmaticum* (Cd1)	*M. dalmatica*		Mt Lovćen (Montenegro)		HFK-HR-51073		1424	
*C. dalmaticum* (Cd2)	*M. dalmatica*		Mt Orjen (Montenegro)		HFK-HR-51051		1074	
*C. dalmaticum* (CdB1)	*M. bulgarica*		Uhlovitsa cave (Bulgaria)		HFK-HR-51076		1040	
*C. dalmaticum* (CdB2)	*M. bulgarica*		Mesta River Valley (Bulgaria)		HFK-HR-51134		580	
*C. dalmaticum* (CdB3)	*M. bulgarica*		Vlahina Mt (Bulgaria)		HFK-HR-51135		1140	
*C. frivaldszkyanum* (Cf1)	*M. frivaldszkyana*		Ostrusha peak (Bulgaria)		HFK-HR-51093		1405	
*C. frivaldszkyanum* (Cf2)	*M. frivaldszkyana*		Vikanata Skala Nature Monument (Bulgaria)		HFK-HR-51137		1040	
*C. pulegium* (Cp)	*M. pulegium*		Međeđa (BIH)		HFK-HR-51050		571	
*C. serpyllifolium* (Cs)	*M. albanica*		Prizren (Kosovo)		HFK-HR-51074		367	
*C. thymifolium* (Ct1)	*M. thymifolia*		Učka Mt (Croatia)		HFK-HR-51077		1189	
*C. thymifolium* (Ct2)	*M. thymifolia*		Dokozina plan (Croatia)		HFK-HR-51082		1441	
*C. thymifolium* (Ct3)	*M. thymifolia*		Šušanj (Croatia)		HFK-HR-51081		604	
*C. thymifolium* (Ct4)	*M. thymifolia*		Panos (Croatia)		HFK-HR-51084		1148	
*C. thymifolium* (Ct5)	*M. thymifolia*		Blidinje (BIH)		HFK-HR-51042		1195	
*C. thymifolium* (Ct6)	*M. thymifolia*		Diva Grabovica (BIH)		HFK-HR-51086		252	
*C. thymifolium* (Ct7)	*M. thymifolia*		Manastir Morača (Montenegro)		HFK-HR-51053		301	
**Taxa (Code)**	**Latitude (N)**	**Longitude (E)**	**n**	** *p* ** **(%)**	** *N* ** **pr**	** *I* **	**DW**	** *H* ** ** _E_ ** ** (*F*_IS_ = 0)**
*M. cristata* (Mc1)	43°22′45.0″	21°53′50.1″	9	25.68	1	0.185	2445.84	0.093
*M. cristata* (Mc2)	42°29′33.4″	23°11′43.1″	7	26.92	1	0.204	3253.74	0.109
*M. cristata* (Mc3)	41°24′18.1″	22°15′47.0″	12	41.68	2	0.287	3606.54	0.118
*M. cristata* (Mc4)	41°15′10.3″	23°24′49.8″	8	34.65	2	0.256	2785.33	0.117
*M. cristata* (Mc5)	40°57′21.2″	22°20′02.0″	3	19.54	0	0.179	2700.39	0.122
*M. cristata* ssp. *kosaninii* (McK)	41°22′09.0″	21°39′06.1″	10	41.56	0	0.295	4153.29	0.130
*M. croatica* (Mcr1)	44°45′51.1″	4°59′17.1″	5	14.58	0	0.117	1242.40	0.073
*M. croatica* (Mcr2)	44°34′45.2″	15°05′49.5″	5	14.70	0	0.118	1139.31	0.075
*M. croatica* (Mcr3)	44°32′37.5″	15°10′04.7″	5	16.47	0	0.133	1311.92	0.083
*M. croatica* (Mcr4)	44°20′57.7″	15°24′50.3″	5	18.95	0	0.155	1508.70	0.090
*M. croatica* (Mcr5)	44°15′17.8″	15°48′58.2″	5	14.34	0	0.115	1237.31	0.073
*M. croatica* (Mcr6)	43°35′59.0″	17°41′04.3″	5	18.54	0	0.155	1357.73	0.090
*M. croatica* (Mcr7)	43°35′10.1″	18°04′44.0″	4	15.76	0	0.135	1540.99	0.085
*M. croatica* (Mcr8)	43°34′10.1″	18°05′59.2″	5	18.06	0	0.147	1232.93	0.083
*M. croatica* (Mcr9)	43°30′21.1″	18°06′20.3″	5	17.77	0	0.143	1379.19	0.084
*M. croatica* (Mcr10)	42°52′28.2″	19°23′05.1″	5	19.48	0	0.162	2049.82	0.095
*M. croatica* (McrP1)	42°57′01.5″	17°21′52.2″	10	25.86	1	0.178	2318.02	0.085
*M. croatica* (McrP2)	42°49′28.1″	17°40′23.7″	10	29.93	1	0.207	3004.68	0.097
*M. graeca* (Mg1)	43°03′43.3″	16°12′55.9″	4	18.00	0	0.154	4188.99	0.108
*M. graeca* (Mg2)	43°02′17.4″	16°05′49.5″	5	19.48	0	0.158	4575.80	0.105
*M. graeca* ssp*. fruticulosa* (MgF)	43°02′13.4″	16°05′50.5″	11	28.34	4	0.195	6622.21	0.099
*M. juliana* (Mj1)	42°26′40.4″	13°36′90.7″	10	30.64	0	0.218	1840.88	0.112
*M. juliana* (Mj2)	42°32′53.9″	18°22′02.3″	9	33.23	0	0.234	1617.68	0.112
*M. juliana* (Mj3)	42°41′45.8″	18°29′26.9″	10	24.50	0	0.171	1287.44	0.091
*M. juliana* (Mj4)	42°24′26.0″	18°47′15.9″	5	23.44	0	0.190	1584.60	0.117
*M. juliana* (Mj5)	42°29′47.8″	18°38′57.1″	5	20.43	0	0.162	1386.40	0.103
*M. juliana* (Mj6)	42°25′44.6″	19°28′53.5″	8	38.61	1	0.287	2257.51	0.131
*M. juliana* (Mj7)	41°41′02.6″	21°48′11.7″	10	38.96	0	0.273	1685.18	0.123
*M. juliana* (Mj8)	41°26′12.2″	21°52′06.4″	8	31.29	1	0.240	2009.83	0.117
*M. juliana* (Mj9)	40°27′30.0″	23°31′04.6″	10	35.01	1	0.246	2766.48	0.122
*M. juliana* (Mj10)	41°16′10.8″	23°25′06.7″	9	31.17	0	0.227	1691.23	0.116
*M. kerneri* (Mk1)	44°42′02.2″	14°54′45.6″	10	25.44	0	0.177	2007.33	0.099
*M. kerneri* (Mk2)	44°17′35.1″	15°26′35.1″	8	26.92	0	0.200	1598.30	0.108
*M. kerneri* (Mk3)	43°45′55.9″	15°59′12.1″	9	26.39	0	0.198	1352.29	0.103
*M. kerneri* (Mk4)	42°57′04.5″	17°05′56.1″	8	23.61	0	0.171	1409.70	0.095
*M. kerneri* (Mk5)	43°20′36.6″	17°48′37.9″	10	20.72	0	0.139	1489.38	0.079
*M. longipedunculata* (Ml1)	42°42′15.9″	18°30′33.3″	10	24.85	0	0.166	1668.95	0.081
*M. longipedunculata* (Ml2)	42°32′46.2″	18°42′35.4″	5	16.94	1	0.137	2381.71	0.085
*M. longipedunculata* (Ml3)	42°46′02.7″	18°57′24.0″	10	24.50	0	0.166	1503.14	0.082
*M. longipedunculata* (Ml4)	42°25′44.6″	19°28′53.5″	10	29.46	0	0.196	1736.85	0.088
*M. microphylla* (Mm)	43°01′07.8″	15°45′08.2″	9	34.30	0	0.239	2097.51	0.120
*C. dalmaticum* (Cd1)	42°23′45.6″	18°50′09.5″	4	19.42	0	0.168	2225.44	0.106
*C. dalmaticum* (Cd2)	42°33′45.1″	18°37′36.6″	4	18.77	0	0.160	3159.10	0.107
*C. dalmaticum* (CdB1)	41°30′49.9″	24°39′35.2″	9	35.48	1	0.248	2764.17	0.113
*C. dalmaticum* (CdB2)	41°40′46.6″	23°43′29.7″	9	36.30	1	0.255	2861.36	0.114
*C. dalmaticum* (CdB3)	41°50′30.6″	22°59′27.8″	8	33.53	0	0.243	2693.05	0.116
*C. frivaldszkyanum* (Cf1)	42°43′54.7″	25°15′53.9″	8	26.33	0	0.190	1953.05	0.100
*C. frivaldszkyanum* (Cf2)	42°45′57.6″	25°30′08.1″	9	29.75	0	0.210	2592.09	0.104
*C. pulegium* (Cp)	43°43′57.6″	19°11′10.9″	3	13.75	0	0.126	2032.69	0.101
*C. serpyllifolium* (Cs)	42°11′52.1″	20°46′11.2″	3	19.78	0	0.182	2514.92	0.123
*C. thymifolium* (Ct1)	45°17′08.1″	14°12′02.4″	9	26.68	0	0.193	1632.85	0.095
*C. thymifolium* (Ct2)	44°39′04.3″	15°02′39.1″	10	24.73	0	0.168	1472.09	0.085
*C. thymifolium* (Ct3)	44°31′33.8″	15°06′45.1″	9	27.92	0	0.198	1787.01	0.096
*C. thymifolium* (Ct4)	44°26′06.1″	15°16′35.2″	10	27.80	0	0.195	1415.32	0.094
*C. thymifolium* (Ct5)	43°31′07.8″	17°23′09.8″	10	22.02	0	0.156	1719.07	0.081
*C. thymifolium* (Ct6)	43°35′59.0″	17°41′04.3″	9	24.73	0	0.176	1522.22	0.090
*C. thymifolium* (Ct7)	42°45′50.9″	19°23′34.6″	9	23.79	1	0.169	1985.02	0.088

Note: *n =* sample size; *p* (%) = proportion of polymorphic bands; *N*pr = number of private bands; *I* = Shannon’s information index; DW = frequency down-weighted marker values; *H*_E_ = expected heterozygosity of population; BIH = Bosnia and Herzegovina; N. = North.

**Table 2 plants-11-03407-t002:** Results of analysis of molecular variance (AMOVA) within taxa.

Taxon Code	Number of Populations	Population Variation (%)		Φ	*p* (Φ)
		Among	Within		
Mc	5	24.82	75.18	0.248	<0.0001
McK	1	–	–	–	–
Mcr	10	14.37	85.63	0.144	<0.0001
McrP	2	12.97	87.03	0.130	<0.0001
Mg	2	5.04	94.96	0.050	0.1184
MgF	1	–	–	–	–
Mj	10	15.99	84.01	0.160	<0.0001
Mk	5	34.55	65.55	0.344	<0.0001
Ml	4	20.44	79.56	0.204	<0.0001
Mm	1	–	–	–	–
Cd	2	11.39	88.61	0.114	0.0293
CdB	3	13.26	86.74	0.133	<0.0001
Cf	2	7.75	92.25	0.077	<0.0002
Cp	1	–	–	–	–
Cs	1	–	–	–	–
Ct	7	18.03	81.97	0.180	<0.0001

Mc, *M. cristata* ssp. *cristata*; McK, *M. cristata* ssp. *kosaninii*; Mcr, *M. croatica* including the former *M. pseudocroatica*, McrP; Mg, *M. graeca* ssp. *graeca*; MgF, *M. graeca* ssp. *fruticulosa*; Mj, *M. juliana*; Mk, *M. kerneri*; Ml, *M. longipedunculata*; Mm, *M. microphylla*; Cd, *C. dalmaticum* including the former *M. bulgarica*, CdB; Cf, *C. frivaldszkyanum*; Cp, *C. pulegium*; Cs, *C. serpyllifolium*; Ct, *C. thymifolium*.

**Table 3 plants-11-03407-t003:** Results of the analysis of molecular variance (AMOVA) between taxa.

Analysis Between	Source of Variation	df	Variance Components	Percent of Total Variance	Φ	*p* (Φ)
	Between taxa	1	1.247	0.87	0.009	0.336
Mc and McK	Among populations within taxa	4	33.494	23.49	0.237	0.0001
	Within populations	43	107.859	75.64	0.244	0.0001
	Between taxa	1	25.412	16.76	0.168	0.0001
Mc1,2,4 and	Among populations within taxa	4	18.385	12.12	0.146	0.0001
Mc3,5, McK	Within populations	43	107.859	71.12	0.289	0.0001
	Between taxa	1	42.040	33.49	0.335	0.0001
Mcr and McrP	Among populations within taxa	10	11.098	8.84	0.133	0.0001
	Within populations	57	72.401	57.67	0.423	0.0001
	Between taxa	1	103.963	55.11	0.551	0.0001
Mg and MgF	Among populations within taxa	1	4.010	2.13	0.047	0.0001
	Within populations	17	80.671	42.76	0.572	0.0001
	Between taxa	1	2.174	1.79	0.018	0.0001
Mk1–5 and Mm	Among populations within taxa	4	39.215	32.32	0.329	0.0001
	Within populations	48	79.927	65.88	0.341	0.0001
	Between taxa	1	10.966	9.76	0.098	0.197
Mk1,2,4,5 and	Among populations within taxa	3	22.807	20.29	0.225	0.0001
Mm	Within populations	40	78.629	69.95	0.300	0.0001
	Between taxa	1	26.066	18.22	0.182	0.002
Mj and Mk1–5	Among population within taxa	13	25.796	18.03	0.220	0.0001
	Within populations	114	91.237	63.76	0.362	0.0001
	Between taxa	1	42.786	27.66	0.277	0.0001
Mj and Mk1,2,4,5	Among populations within taxa	12	20.317	13.13	0.181	0.0001
	Within populations	106	91.601	59.21	0.408	0.0001
	Between taxa	1	33.181	21.91	0.219	0.0001
Cd and CdB	Among populations within taxa	3	15.084	9.96	0.128	0.0001
	Within populations	29	103.201	68.13	0.319	0.0001
	Between taxa	1	27.520	19.62	0.196	0.0001
CdB and Cf	Among populations within taxa	3	13.551	9.66	0.120	0.0001
	Within populations	38	99.228	70.73	0.293	0.0001
	Between taxa	1	45.458	32.20	0.322	0.336
Cd and Cf	Among populations within taxa	2	8.599	6.09	0.090	0.0001
	Within populations	21	87.135	61.71	0.383	0.0001

Mc, *M. cristata* ssp. *cristata*; McK, *M. cristata* ssp. *kosaninii*; Mcr, *M. croatica* including the former *M. pseudocroatica*, McrP; Mg, *M. graeca* ssp. *graeca*; MgF, *M. graeca* ssp. *fruticulosa*; Mj, *M. juliana*; Mk, *M. kerneri*; Ml, *M. longipedunculata*; Mm, *M. microphylla*; Cd, *C. dalmaticum* including the former *M. bulgarica*, CdB; Cf, *C. frivaldszkyanum*; Cp, *C. pulegium*; Cs, *C. serpyllifolium*; Ct, *C. thymifolium*.

**Table 4 plants-11-03407-t004:** Pearson’s correlation coefficients between 21 main compounds and scores of the first two principal components (PC).

Compound	PC1		PC2	
*α*-Pinene	0.237	ns	−0.010	ns
*β*-Pinene	0.275	*	0.059	ns
*α*-Campholenal	0.083	ns	0.157	ns
Menthone	−0.446	***	−0.182	ns
Camphor	0.264	*	0.692	***
Pinocarvone	−0.186	ns	0.191	ns
Borneol	0.364	**	0.814	***
Verbenone	0.654	***	−0.558	***
Pulegone	−0.744	***	−0.245	ns
Piperitonene	−0.360	**	0.142	ns
Piperitone oxide	−0.484	***	−0.151	ns
*E*-Caryophyllene	0.625	***	−0.194	ns
Germacrene D	0.463	***	0.479	***
Spathulenol	−0.290	*	−0.037	ns
Caryophyllene oxide	0.836	***	−0.335	*
*α*-Muurolol	−0.041	ns	−0.130	ns
*α*-Cadinol	0.105	ns	0.305	*
*α*-Bisabolol	−0.084	ns	0.122	ns
Eudesm-7(11)-en-4-ol	−0.023	ns	−0.081	ns
Thymol	0.278	*	−0.330	*
Docosane	0.250	ns	−0.635	***
Eigenvalue	3.468		2.677	
% of variance	16.51		12.75	

ns = non-significant; * = significant at *p* < 0.05; ** = significant at *p* < 0.01; *** = significant at *p* < 0.001.

## Data Availability

Not applicable.

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
