# Peer review of "Phytochemicals and Their Correlation with Molecular Data in Micromeria and Clinopodium (Lamiaceae) Taxa"

_plants, 2022, doi:10.3390/plants11233407_

Round 1

Reviewer 1 Report

This is a very interesting work because it brings together the morphological systematics, DNA sequences and phytochemistry of the essential oils of Micromeria and Clinopodium in the Balkan Peninsula. There are few works that propose a comprehensive treatment of a group of species.

I would add to the beginning of the introduction some general works that include phytochemistry as a taxonomic character. For example, Judd. Plant Systematics. A phylogenetic approach (chapter 4), 2018; Gershenzon & Mabry. Secondary metabolites and the higher classification of angiosperms. Nordic Journal of Botany 1983, 3, 5-34; Crawford. Flavonoid chemistry and angiosperm evolution. The Botanical Review 1978, 44, 431-456, etc.

The manuscript has the added value of discussing some synonymisations of taxa, including biogeographical comments, as well as comparing its results with those from other areas of Europe, such as the Iberian Peninsula.

Some editing aspects:

1- Take care that whenever the name of a taxon is cited for the first time in a paragraph, the genus is not abbreviated. E.g. L124, L143, etc.

2- Take care that all taxon names are written in italics. Also in the captions. For example, in Table 2 or Figure 4.

3- L269: Change ‘1. Discussion’ to ‘3. Discussion’

Author Response

Dario Kremer

University of Zagreb

Faculty of Pharmacy and Biochemistry

A. Kovačića 1,

10000 Zagreb

Croatia

e-mail: kremer@pharma.hr

Tel.: + 38-51-4619-422

Zagreb, November, 30. 2022

            Dear Reviewer,

 Please find enclosed answers on your comments on manuscript 'Phytochemicals and their correlation with molecular data in Micromeria and Clinopodium (Lamiaceae)’ taxa after review.

 List of comments are given below.

 Comment: I would add to the beginning of the introduction some general works that include phytochemistry as a taxonomic character. For example, Judd. Plant Systematics. A phylogenetic approach (chapter 4), 2018; Gershenzon & Mabry. Secondary metabolites and the higher classification of angiosperms. Nordic Journal of Botany 1983, 3, 5-34; Crawford. Flavonoid chemistry and angiosperm evolution. The Botanical Review 1978, 44, 431-456, etc.

Author response: The comment is accepted and the proposed references have been added to the manuscript in the Introduction section (paragraph 3), where the authors mention chemical investigations in resolving taxonomic issues.

 Comment: Take care that whenever the name of a taxon is cited for the first time in a paragraph, the genus is not abbreviated. E.g. L124, L143, etc.

Author response: The comment is accepted.

 Comment: Take care that all taxon names are written in italics. Also in the captions. For example, in Table 2 or Figure 4.

Author response: The comment is accepted.

 Comment:  3- L269: Change ‘1. Discussion’ to ‘3. Discussion’

Author response: The comment is accepted. 

Sincerely, Dario Kremer

Reviewer 2 Report

Title: Phytochemicals and their correlation with molecular data in Micromeria and Clinopodium (Lamiaceae) taxa

Authors: Dario Kremer, Valerija Dunkić, Ivan Radosavljević, Faruk Bogunić, Daniella Ivanova, Dalibor Ballian, Danijela Stešević, Vlado Matevski, Vladimir Ranđelović, Eleni Eleftheriadou, Zlatko Šatović, and Zlatko Liber

Reference: plants-2066976

Article type: Research

Reviewer Comments:

The manuscript plants-2066976, entitled “Phytochemicals and their correlation with molecular data in Micromeria and Clinopodium (Lamiaceae) taxa”, studies the phytochemical and molecular characteristics of Micromeria and Clinopodium taxa (family Lamiaceae) distributed in the Balkan Peninsula.

Despite the lack of innovative features, the topic covered in the present papers is important and targets a broad range of readers presenting implications in several fields of knowledge.

General comments:

In the reviewer's opinion, the authors of this paper do not have English as their native tongue. A high degree of editing and revising will be necessary. Also, some aspects concerning assertiveness (e.g., avoi expressions such as “could be explained”), scientific accuracy (e.g., weak, but significant correlation – if the correlation is not significant, it should not be reported; please consider removing “but significant”) have to be improved, and text clarity.

Due to the extension of the corrections needed, instead of presenting the suggestion by line, the reviewer presents the bulk sections of the text (suggestions in bold).

The Abstract is comprehensive and well-structured.

The Introduction section is quite confusing and should be improved for the simplification and clarity of the text.

The Result section is deeply confusing and should be improved for the simplification and clarity of the text.

The Discussion section is deeply confusing and should be improved for the simplification and clarity of the text.

The Material and Methods needs to be simplified. If the ordinary protocols were used, there is no need to describe them. It is enough to cite the reference and point out if modifications were made.

The Conclusion section covers general concepts. An effort should be made to present conclusion related to the presented work.

Specific comments:

Abstract: please consider replacing it with

A study of the phytochemical and molecular characteristics of ten Micromeria and six Clinopodium taxa (family Lamiaceae) distributed in the Balkan Peninsula was carried out. The phytochemicals detected in essential oils (EO) by gas chromatography, mass spectrometry, and molecular data (amplified fragment length polymorphism, AFLP) were used to study the taxonomic relationships between the taxa and the correlations between phytochemical and molecular data. The STRUCTURE analysis revealed three genetic groups; while the Bayesian Analysis of Population Structure (BAPS) grouped the studied taxa into 11 clusters nested in the groups obtained by STRUCTURE. The PCA analysis performed with the 21 most represented compounds in EO yielded results partially consistent with those obtained by STRUCTURE and neighbour-joining. Their geographic distributions did not support the genetic grouping of the studied taxa and populations. Mantel test showed a weak correlation between phytochemical and genetic data (r = 0.421, p=0.001). The 17.8% of the phytochemical distance between populations is explained by genetic distance. The current taxonomic position of some studied taxa has yet to be satisfactorily resolved, and further studies are needed. Such future research should include nuclear and plastid DNA sequences from a more significant number of populations and individuals.

Introduction: please consider replacing it with

The genus Micromeria Benth. (Lamiaceae) includes 54 [1] or 70/20 [2] annual and 44 perennial herbs, sub-shrubs, and shrubs, depending on the point of view. According to Bräuchler et al. [1,3], the distribution of Micromeria species extends from the Mediterranean to South Africa and Madagascar and from China to the Macaronesian Archipelago. Chater and Guinea [4] described Micromeria species for Europe, with more than half of these species occurring in the Balkan Peninsula. The genus Micromeria is part of a complex group of genera in the tribe Mentheae and subtribe Menthinae (subfamily Nepetoideae, family Lamiaceae) and was often considered part of the vaguely defined "Satureja" complex [5–7]. On the other hand, Bentham [8] accepted Micromeria species strictly as a distinct genus, and this opinion prevails today [9–12]. Harley et al. [13] proposed four sections (Micromeria, Pineolentia, Cymularia, and Pseudomelissa) concerning the infrageneric subdivision of the genus Micromeria, while Doroszenko [10] described the morphological traits of these sections. The species of genus Micromeria inhabiting the Balkan Peninsula, which are the focus of this article, belong to the sections Pseudomelissa and Micromeria [13] or Pseudomelissa and Eumicromeria [14]. Bräuchler et al. [3] showed that the genus Micromeria is polyphyletic and that a revision of this genus is necessary. Based on this view, Bräuchler et al. [15] transferred the section Pseudomelissa from the genus Micromeria to the genus Clinopodium L. Bräuchler et al. [1] also provided a comprehensive list of new combinations, synonyms, and valid names. Studies on micromorphological characters of Balkan Micromeria and closely related Clinopodium species support the recent transfer of section Pseudomelissa to the genus Clinopodium [16].

The genus Micromeria is represented in the Balkan Peninsula by several species with narrow distribution, which are primarily included on local floras and lists [17–23]. Their taxonomic position, based on the morphological characters, was sometimes not clearly defined, followed by variable, complex synonymy, and the author's subjectivism. Despite a new taxonomic position of section Pseudomelissa, many authors [24–30] still use the previously assigned names.

Genetic studies can help reconstruct the evolutionary history and the delimitation of species or subspecies [31–32]. The evolutionary history and underlying genetic structure of closely related taxa may be confronted with different habitats that impose particular environmental constraints on them [33]. Although genetic analyses provide the most helpful information for taxonomic studies today, chemical investigations such as the detection of essential oils (EO) or phenolic substances can also help solve taxonomic problems [34–36]. Several authors [24–25, 27, 30, 37–39] investigated the essential oil content in Micromeria and Clinopodium species. However, only a few of them [34,37] tried to find connections between EO content and the species' taxonomic positions.

This study aims to obtain additional knowledge about Micromeria and closely related Clinopodium species recently transferred from the section Pseudomelissa and widely distributed in the Balkan Peninsula. To achieve this goal, molecular and phytochemical studies were performed at the population level on 16 Balkan Micromeria and closely related Clinopodium taxa.

Results: please consider replacing it with

2.1. AFLP Analysis

The AFLP analysis revealed high contrasts in population-genetic parameters among studied populations. The percentage of polymorphic fragments and Shannon’s index (Table 1) varied among the studied taxa, with the lowest values observed in C. pulegium (Cp) (13.75%; 0.126) and the highest in M. cristata ssp. cristata (population Mc3) (41.68%; 0.287). Out of 1694 polymorphic markers in 434 individuals, 19 were private. They were detected across 14 populations, most of which belonged to populations of M. cristata ssp. cristata (six private alleles) and a single population of M. graeca ssp. fruticulosa (four private alleles), whereas no private alleles were detected in populations of M. kerneri. Of the 16 Clinopodium populations, private alleles were detected in only three. Frequency down-weighted marker values (DW) ranged from 1139.31 (M. croatica, population Mcr2) to 6622.21 (M. graeca ssp. fruticulosa). The highest values were detected in M. graeca and M. cristata, while much lower values were found in the other studied taxa. The expected heterozygosity (HE) levels ranged from 0.073 (populations Mcr1 and Mcr5 of M. croatica) to 0.130 (M. cristata ssp. kosaninii, McrK) and 0.131 (population Mj6 of M. juliana). With an average value of 0.114, M. cristata and M. juliana had the highest HE levels, while on the other side of the spectrum were M. longipedunculata and M. croatica, with an average HE value of 0.084. Of the total variability, 48.68% refers to the variability between and 51.32% within populations, indicating significant differences between the studied populations.

AMOVA analysis showed that intrapopulation variability was considerably higher than among populations (Table 2). Variability among populations ranged from 5.04% (M. graeca ssp. graeca) to 34.45% (M. kerneri). For most species, interpopulation variability was approximately 10%. Exceptions were the populations of M. cristata (24.82%) and M. kerneri (34.45%) with a higher interpopulation distance (ΦST).

The results of neighbour-joining (NJ) analysis are shown in Figure 1. Three genetic groups determined by STRUCTURE were marked by the same colours (blue, green, red) in the NJ tree to allow comparison of the two analyses. The NJ analysis confirms with significant bootstrap support that Micromeria and Clinopodium are well-differentiated groups of closely related taxa. In addition, most of the individuals in both groups were well supported, except for the M. juliana – M. kerneri – M. microphylla cluster, characterized by low differentiation among the adopted taxa. The individuals of M. juliana were grouped in the same cluster with M. kerneri from the Croatian population Gradina (Mk3). The population of M. microphylla (Mm) was divided into two clusters and associated with four populations of M. kerneriAMOVA results showed no statistically significant differences between M. microphylla and M. kerneri

Two populations of M. croatica, known from local Balkan literature as M. pseudocroatica (McrP), were separated from the other ten populations of M. croatica (Mcr) (Figure 1, Table 3). Also, the population of M. graeca ssp. fruticulosa (MgF) was recognized as a differentiated taxon, separated from the two populations of M. graeca ssp. graeca (Mg). In the NJ tree, the population of M. cristata ssp. kosaninii (McK) and two populations of M. cristata ssp. cristata (Mc3, Mc5) were also separated.

The position of the studied taxa within the genus Clinopodium is complex. Clinopodium dalmaticum showed genetic differentiation into two subgroups. The first subgroup was formed by two Montenegrin populations (Cd), while the second was formed by three Bulgarian populations (CdB). Moreover, the Bulgarian populations are slightly closer to C. frivaldszkyanum (Cf) than the Montenegrin populations of the same species (Figure 1). Three other Clinopodium species studied (C. serpyllifoliumC. pulegium, and C. thymifoliumwere separated from the other taxa with high bootstrap support. 

In the STRUCTURE analysis the highest K value was observed for K = 3 (K = 1381394.43), indicating the presence of three genetic clusters, which was considerably larger than all other K values (Figure S1). The STRUCTURE analysis (Figure 2A) revealed three genetic groups, shown as tree colours (blue, green, and red) on the NJ tree (Figure 1). The first group included populations of M. cristataM. croaticaM. graeca, and M. longipedunculata; the second included the populations of M. juliana and M. kerneri; the third included all Clinopodium populations studied. While low levels of admixture characterized most of the studied populations, this was not the case with the M. microphylla population that was positioned between the two Micromeria clusters with high admixture levels. The genetic clusters were not geographically defined since representatives of each cluster were found throughout the study. 

On the other hand, BAPS analysis (Figure 2B) revealed a congruent assignment of the studied Micromeria and Clinopodium taxa to 11 clusters nested within groups identified by STRUCTURE analysis. The best partitions received log-likelihoods of –182699.06 at P = 1 (without using geographic coordinates as informative priors) and – 183284.74 at P = 1 (spatial clustering). In general, both methods produced nearly identical results. The first two groups of the BAPS analysis were formed by M. cristata ssp. cristata (Mc) and M. cristata ssp. kosaninii (McK); the third group by M. croatica (Mcr); the fourth by two populations of M. croatica covered under the disputed name M. pseudocroatica (McrP); the fifth and the sixth clusters were formed by M. graeca ssp. graeca (Mg) and M. graeca ssp. fruticulosa (MgF), respectively; the seventh by M. juliana (Mj) and one population of M. kerneri (Mk3); the eighth by M. microphylla (Mm) and four populations of M. kerneri (Mk1–Mk3, Mk5); the ninth by M. longipedunculata (Ml); the tenth by C. pulegium (Cp), C. serpyllifolium (Cs), C. dalmaticum (Cd), and C. thymifolium populations; the eleventh cluster was formed by C. frivaldszkyanum (Cf) and three C. dalmaticum (CdB), listed under the controversial name M. bulgarica.

2.2. Essential Oil Analysis

The composition and yield of EO included 41 oil samples from the genus Micromeria and 15 representatives from the genus Clinopodium (Tables S1–S8). The yields of the studied taxa ranged from a minimum of 0.35% in C. frivaldszkyanum to 1.79% in M. croatica. The composition of EO of all studied taxa can be divided into the following classes: monoterpene hydrocarbons (1.66–44.73%), oxygenated monoterpenes (10.42–85.31%), sesquiterpene hydrocarbons (1.09–35.73%), oxygenated sesquiterpenes (0.25–39.39%), phenolic compounds (0–25.73%, carbonyl compounds (0–2.56%), and hydrocarbons (0.25–14.07%). 

Further presentation of the results focuses on the main volatile components in the composition of the EO of the studied species. In the EO composition of the studied five populations of M. cristata ssp. cristata, the compounds borneol (14.11–26.28%) and α-cadinol (12.48–17.72%) were the most abundant. The compounds α-muurolol (17.53%) and pulegone (8.91%) were detected only in M. cristata subsp. kosaninii. Other notable compositional differences are the detection of verbenone, camphene, bornyl acetate, and α-humulene in all populations of M. cristata but not in M. cristata subsp. kosaninii (Table S1). The compounds borneol (16.13–28.71%), E-caryophyllene (7.13–16.8%), caryophyllene oxide (10.92–15.75%), and germacrene D (2.95–14.12%) were the main components of the EO extracted from the 12 samples of M. croatica examined (Tables S2 and S3).

In the EO of M. graeca ssp. graeca, α-bisabolol was present in a very high percentage at both studied sites (23.02% and 25.78%). In the EO of M. graeca subsp. fruticulosa, α-bisabolol was also abundant (11.92%), but the most representative compound was pinocarvone (17.39%) (Table S3).

In the composition of the ten samples of EO of M. juliana, the most abundant compounds were E-caryophyllene (10.62–22.35%) and caryophyllene oxide (22.26–32.72%) (Table S4). Also, in the five studied EO of M. kerneri, caryophyllene oxide (12.81–23.46%) was the most abundant compound, followed by α-pinene (12.3–16.13%) (Table S5).

Isolates of the species M. longipedunculata contained the most spathulenol (more than 30% of all four populations studied), while M. microphylla was rich in eudesem-7 (11)-en-4-ol (22.91%) (Table S6).

Clinopodium dalmaticum showed the most significant differences in EO composition among investigated Clinopodium taxa. The oil composition in the Montenegro samples consisted mainly of piperitone (more than 30%). Two Bulgarian populations presented high concentrations of E-caryophyllene (CdB1, 31.74%; CdB2, 42.43%), while population CdB3 had the highest content of α-pinene (14.31%). The compound thymol was also significantly present in Bulgarian populations of C. dalmaticum. Pulegone (Cf1, 47.2%; Cf2, 29.52%) and menthone (Cf1, 12.8%; Cf2, 9.23%) predominate in the composition of the Bulgarian species of C. frivaldszkyanum (Table S7).

In all other Clinopodium species studied (C. pulegiumC. serpyllifolium, and C. thymifolium), the most abundant compounds were pulegone and piperitenone oxide, making these oils extremely rich in oxygenated monoterpenes (64.99–85.31%( (Table S8). 

PCA analysis (Figure 3) was performed on the 21 compounds isolated from the EO, which is greater than 10% in one sample (population). PC1 and PC2 for compounds of EO explained 29.26% of the variance. The phytochemical groups obtained by PCA are partly similar to the three genetic groups determined by NJ and STRUCTURE. Among the 21 compounds, PCA detected nine components that contribute most to the differences between groups (Figure 3)Clinopodium species are mainly located in the negative region of PC1 and PC2. M. longipedunculata (Ml) is located among the Clinopodium species.

The main compounds of this group were menthone, pulegone, and piperitone oxide. The highest values of these compounds were found in C. frivaldszkyanum (Cf1, menthone) and C. thymifolium (Ct1, pulegone; Ct7, piperitone oxide) (Tables S7, S8). Only C. dalmaticum from Bulgaria (formerly M. bulgarica, CdB) had an unusual position among the Clinopodium species, being located together with M. kerneri and M. juliana in the negative region of PC1 and the positive region of PC2. The specific compounds for this phytochemical group were verbenone, caryophyllene oxide, and docosane, whose highest values were determined in three populations of M. juliana (Mj7, Mj8, Mj5)(Table S4).

Populations belonging to the third phytochemical group were primarily located in the positive region of PC1 and PC2 (Figure 3). Camphor, borneol, and germacrene D are components that distinguished this group. The highest values of these compounds were determined in M. croatica (Mcr10, germacrene D) and debatable taxon M. pseudocroatica (McrP1, camphor; McrP2, borneol) (Tables S2, S3). Populations belonging to the third phytochemical group were primarily located in the positive region of PC1 and PC2 (Figure 3). Camphor, borneol, and germacrene D are components that distinguished this group. The highest values of these compounds were determined in M. croatica (Mcr10, germacrene D) and debatable taxon M. pseudocroatica (McrP1, camphor; McrP2, borneol) (Tables S2, S3). 

2.3. Mantel test

The correlations between AFLP and phytochemical matrices of dissimilarity were calculated using the Mantel test. A weak correlation was observed between phytochemical and molecular data (r = 0.421, pMantel 0.001). According to the same test, 17.8% (R = 0.178) of the phytochemical distance between populations could be explained by

genetic distance (Figure S2)

Discussion: please consider replacing it with

3. Discussion

Several phylogenetic conclusions can be drawn from genetic diversity and STRUCTURE results. The NJ analysis separated Micromeria from Clinopodium taxa. It reinforced the recent transfer of species from the section Pseudomelissa (genus Micromeria) to the genus Clinopodium by Bräuchler et al. [15]. Although the distinction between the Micromeria and Clinopodium groups was confirmed, it is questionable whether it is substantial enough to label these groups as separate genera. The STRUCTURE analysis results suggest the existence of three genetic clusters: two within Micromeria and a third comprising Clinopodium species. If Clinopodium is treated as a separate genus, the other two Micromeria groups should also be considered as such. Because the previous Clinopodium – Micromeria segregation was based on a reduced number of samples and analysis of a single cpDNA region [3], the result presented here is even more relevant. The NJ analysis showed genetic differentiation of C. dalmaticum in the Montenegrin and Bulgarian populations. Regarding the EO composition, PCA also separated the two populations of C. dalmaticum (Figure 3). These results suggest variability within C. dalmaticum, being Bulgarian populations previously considered to be M. bulgarica [19–20]. Variability within C. dalmaticum was previously described by Vandas [40], which identified M. bulgarica in the area of Usunža and Krivska River (North Macedonia). Chater and Guinea [4] and Ančev [19] identified two subspecies of M. dalmatica (now C. dalmaticum): M. dalmatica ssp. dalmatica and M. dalmatica ssp. bulgarica (Velen.) Guinea. On the other hand, Bräuchler et al. [1,15] concluded that M. bulgarica is a synonym of C. dalmaticumBecause C. dalmaticum showed both molecular and phytochemical separation, research on this species should be continued. The Bulgarian population of C. dalmaticum is closer to C. frivaldszkyanum than the Montenegrin population of C. dalmaticum (Figure 1). The results suggest that the taxonomic relationships within these two or three taxa of the genus Clinopodium should be clarified in the future. Given the considerable geographic distance between the Montenegrin and Bulgarian populations of C. dalmaticumthe existence of two geographically distinct groups is not unusual. Such the refugia-within-refugia model developed by Gómes et Lunt [41] for the Iberian Peninsula has also been applied to some species from the Balkan Peninsula [42–44].

In such studies, which include either individual species or groups of closely related species on the Balkan Peninsula, the question of the presence of (micro)refugial areas that protected local populations during the unfavorable climatic conditions of the glaciation cycles cannot be avoided. In AFLP-based analyses, DW markers (Table 1) are considered indicators for detecting such areas. The results presented here are partially contradictory, making it difficult to draw a plausible conclusion. The highest values of DW markers were observed in M. graeca populations from the islands of the central Adriatic, followed by some M. cristata populations from the eastern and central Balkans (Mc2, Mc3, and McK) and a single population of M. croatica (McrP2) from the southern Adriatic mainland. Populations characterized by moderately high levels of DW markers were more common than (complete). They were also scattered over a large area ranging from central parts of the eastern Adriatic coastal region to the eastern Balkans.

Similar to the previously discussed results from STRUCTURE, no spatial structuring of populations characterized by high frequencies of DW markers can be observed, suggesting that no single refugial area can be identified within the Balkan Peninsula. Instead, there appear to be numerous microrefugia scattered over large areas. The northern Adriatic coastal area does not harbour any of this microrefugia, as low levels of these markers characterized all studied populations.

The NJ analysis separated Micromeria species into several genetic groups and raised questions about the systematic position of some taxa. Populations of M. cristata were separated into two statistically significant different subgroups (Figure 1, Table 3). One subgroup encompassed two western populations of M. cristata ssp. cristata (Mc3, Mc5) and M. cristata ssp. kosaninii (McK), while the second subgroup was formed by three eastern populations of M. cristata ssp. cristata (Mc1, Mc2, Mc4). On the other hand, the PCA analysis based on phytochemical compounds well separated M. cristata ssp. kosaninii from populations of M. cristata ssp. cristata (Figure 3). A possible reason for this is the different conditions of the habitat, which significantly affect the content of the EO [45]. Two populations of M. croatica (McrP1, McrP2) known in Balkan literature [18,23] under the name M. pseudocroatica (McrP) were separated from other ten populations of M. croatica (Mcr1–Mcr10) with high bootstrap support. The PCA analysis of phytochemical traits also split M. croatica (Mcr) from a disputable taxon M. pseudocroatica (McrP). Additionally, the difference between M. croatica and M. pseudocroatica was greater than the difference among the ten populations of M. croatica. Although Bräuchler et al. [1] concluded that M. pseudocroatica is only a synonym of M. croatica, future genetic research should check whether the differences presented herein might be due to geographical isolation. Namely, populations of disputable M. pseudocroatica are located on Pelješac Peninsula and are partly isolated from continental populations of M. croaticaThe population of M. microphylla (Mm) was divided into two clusters associated with four populations of M. kerneri, and there is no statistically significant difference between the two species (Table 3). On the other hand, the PCA analysis of EO compounds showed that M. microphylla (Mm) is quite different from populations of M. kerneri (Figure 3). Different habitat conditions can also explain the differences in essential oil composition between these two species. The additional complexity in this group is contributed by one population of M. kerneri (Mk3), divided into two clusters closely related to M. juliana (Figure 1). The results of AMOVA (Table 3) showed a difference between M. juliana and M. kerneri regardless of the presence of the disputable population Mk3. The PCA analysis of phytochemicals also suggests differences between M. kerneri and M. juliana (Figure 3). On the other hand, Bräuchler et al. [1] consider that M. juliana and M. kerneri might be conspecific. The obtained results indicate molecular and phytochemical differences between M. kerneri and M. julianaThe additional molecular analysis should check whether the Balkan populations of M. kerneri belong to M. juliana and their relationship to the M. microphyllaThe STRUCTURE analysis (Figure 2A) detected a general distribution in three genetic groups. Their geographic distributions did not support the genetic grouping of the studied taxa and populations. Surprisingly, the spatial structure of recognized genetic clusters from STRUCTURE analysis virtually does not exist, as representatives from all groups are found across large regions, mixed in a seemingly random fashion. Such a result is hard to explain, but it confirms that geographic distribution areas of studied taxa do not follow the levels of their relatedness. It should be noted that a similar situation was also detected, where numerous populations of genetically well-supported taxa scattered over large areas without any signs of spatial groupings. Such a result suggests that strong gene flow barriers among closely related taxa are present, eliminating any possibility for inter-species hybridization and consequent fusion of these taxa into spatially structured clusters. The exception is mentioned in the M. juliana – M. kerneri – M. microphylla complex, where these barriers seem weakest, if present. It is impossible to explain these results based on the obtained results. Possibly, it is a consequence of contrasting evolutionary histories and environmental conditions these taxa are experiencing that have resulted in the development of strong reproductive isolation mechanisms. Another set of results enabled a more straightforward conclusion to be made. Not only were the majority of analysed taxa well supported (except for M. juliana – M. kerneri – M. microphylla complex), but also recognition of a few additional taxa is now possible, thus opening the possibility for some systematic repositioning within the studied groups. Within the C. dalmaticum group, two well-supported taxa were identified, one formerly known as M. dalmatica and another as M. bulgarica. Within the Micromeria group, a similar result was observed in three cases. M. graeca ssp. fruticulosa, formerly known as M. fruticulosa, emerged as a well-supported taxon and not a representative of M. graeca. Similarly, two populations of M. croatica, formerly recognized as M. pseudocroatica were differentiated from any other taxon, thus confirming their status as a separate species. Within the M. cristata group, two taxa have emerged: one comprising populations Mc3, Mc5, and McK, and another comprising Mc1, Mc2, and Mc4. M. cristata ssp. kosaninii (McK), formerly recognized as M. kosaninii, seems to lack needed support for its recognition as either species or subspecies. However, bearing in mind that Bräuchler et al. [1] validated a new combination of M. cristata ssp. kosaninii, future research should also check the taxonomic position of this taxon and whether some western populations of M. cristata ssp. cristata belong to a new combination, M. cristata ssp. kosaninii.

It is known that the content of EO depends on the developmental stage of the plant and the collection site [45]. To exclude the influence of the plant's developmental stage, the aboveground plant parts of all investigated taxa were collected for isolation of EO at flowering time. The composition of EO of Micromeria and Clinopodium taxa were investigated in all populations of the studied taxa. In general, the results presented in this study are consistent with the pattern of literature data previously obtained by other authors. 

In the composition of EO of M. cristata collected in Serbia, Bulgaria, Greece, and Northern Macedonia, the most abundant compound was borneol (14.11–26.28%) (Table S1). Kostadinova et al. [46] also identified borneol in the sample of M. cristata from Bosnek (Bulgaria) with 6.1%, while its isomer isoborneol (11.3%) was the most abundant in the sample collected in Serbia [47]. The extent to which subspecies can also differ in the composition of EO was shown by Çarikçi [48], who studied three subspecies of M. cristata, namely M. cristata ssp. cristataM. cristata ssp. phyrigia P. H. Davis, and M. cristata ssp. orientalis P. H. Davis. The main constituents of EO were borneol and camphor in all three subspecies [48]. Thus, the compounds α-muurolol and pulegone were predominant in the taxon M. cristata ssp. kosaninii (Table S1) have not been identified in the subspecies from Turkey. These differences are not surprising, considering the geographical distance and habitat conditions. Borneol was also one of the main compounds in the studied samples of M. croatica (Tables S2 and S3) from Croatia, Bosnia and Herzegovina, and Montenegro, followed by the compounds E-caryophyllene and caryophyllene oxide. Caryophyllene oxide was the main compound in most of the studied populations of M. croatica, according to Slavkovska et al. [34], Kremer et al. [49], and Vuko et al. [50]. The EO of M. graeca ssp. graeca studied in this research was rich in α-bisabolol (Table S3), while in the same taxa from Greece, the main component in EO was epi-α-bisabolol [51]. In the composition of the ten EO of M. juliana, the most abundant volatile components were E-caryophyllene and caryophyllene oxide (Table S4). Similar to the results of this study, these two compounds also dominated the composition of EO of M. juliana from Anatolia, Turkey [48]. Caryophyllene oxide (12.81–23.46%) was the most abundant compound in the five M. kerneri oils studied, and the following dominant compound was α-pinene (12.3–16.13%) (Table S5). The previous study also showed that the composition of the EO of M. kerneri and M. juliana was characterized by a high concentration of oxygenated sesquiterpenes, with caryophyllene oxide being the most critical compound [37]. In this study, the composition of EO of M. microphylla was reported for the first time, dominated by eudesem-7-(11)-en-4-ol (22.91%) (Table S6). The peculiarity of the oil composition is not surprising, considering the isolation of this population in the central Adriatic (Table 1). Clinopodium dalmaticum is an endemic Balkan species widespread in Bulgaria, Greece, Crete, and Montenegro [52]. In this study, the volatile compounds of samples of this species collected in Montenegro and Bulgaria were identified. In the composition of the isolates from Montenegro, the predominant compound was piperitoneand the Bulgarian samples were rich in E-caryophyllene, α-pinene, and thymol (Table S7). The most abundant compounds were studied in four other Clinopodium taxa (C. frivaldszkyanumC. pulegium, C. serpyllifolium, and C. thymifolium). The most abundant compounds were pulegone and piperitenone oxide, making these oils extremely rich in oxygenated monoterpenes, 53–85.31% (Tables S7 and S8). According to Zheljazkov [53], pulegone was one of the main constituents in the EO of C. frivaldszkyanum from the Bulgarian populations of Shipka and Uzana. The Mantel test showed a weak correlation between the phytochemical and AFLP data (r = 0.421, p 0.001). Literature data on this topic are diverse. According to Slavkovska et al. [34], the composition and quantity of EO of Micromeria species distinguished section Pseudomelissa from Eumicromeria. The EO of species from section Pseudomelissa dominated oxygenated monoterpenes of the menthane type, while the species from section Eumicromeria dominated various terpene compounds [34]. Multivariate analysis (PCA and UPGMA) of compositions determined in the EO of M. kerneri and M. juliana separated the populations of these two species [37]. Feulner et al. [36] determined strong and significant correlations between AFLP data and floral scent volatiles at the population level (r = 0.791, p = 0.004*) and at the individual level (r = 0.823, p < 0.001*) in Sorbus taxa (family Rosaceae). Xavier et al. [54] found a significant correlation between volatile chemical classes and genetic traits of Aniba species in the Amazon region in the Pará State (Brazil). Additionally, AFLP profiles of the eleven Hypericum species and cultivars were correlated with their levels of phytochemicals marker (chlorogenic acid, hyperforin, hypericin, pseudohypericin, and rutin) determined in their methanolic extracts enabling true-to-type identification and marker-assisted breeding programs [55]. Investigations of twenty populations of four Thymus L. species native to Hungary found only partial similarities between dendrograms generated by hierarchical cluster analysis based on DNA patterns and EO samples [56]. On the other hand, in a study of Ophrys taxa (Orchidaceae), Stökl et al. [57] did not find any correlation between the scent data and DNA-molecular data. Trindade et al. concluded that there was no correlation between the chemical analysis of EO of Thymus caespititius Brot. from the Azores and molecular data sets [58, 59]. Volatile and molecular analysis of Juniperus brevifolia (Seub.) Antoine from the same archipelago also showed no correlation between chemical and molecular data sets [60]. Finally, Emami-Tabatabaei et al. [61] studied the possible correlation between the AFLP data and the EO profile obtained by GC-MS of Lutea elbursensis Mozaff from northern Iran. They concluded that the chemical composition of EO cannot be used as a reliable taxonomic tool.

Material and Methods: please consider replacing it with

4. Materials and Methods

4.1. Plant material

Randomly selected samples of wild-growing plants of Micromeria and closely related Clinopodium species were collected during the blooming period from June to August 2018. Voucher specimens of herbal material were deposited in the Herbarium "Fran Kušan", Faculty of Pharmacy and Biochemistry, University of Zagreb, Croatia (Table 1, Figure 4). For molecular analysis, several young leaves from 3 to 11 plants per population were collected on a dry day. Immediately after collection, leaves were dried in plastic bags containing silica gel and stored for further use in DNA analysis. Additionally, the above-ground shoots with leaves and flowers were harvested and mixed to obtain a randomly selected sample. The collected plant parts were air-dried and protected from direct sunlight for 15 days at 22º C and 60 % relative air humidity. From each locality, 50 g of the air-dried plant material was hydrodistilled for 3 h in a Clevenger-type apparatus. The EO obtained was dried over anhydrous sodium sulfate.

4.2. Molecular analysis

4.2.1. DNA isolation

Genomic DNA was isolated using a commercial DNA isolation kit (GenElute™ Plant Genomic DNA Miniprep Kit, Sigma-Aldrich®, Germany), while nanophotometer P330 (Implen®, Germany) was used to measure DNA concentrations and quality. The AFLP technique [62] was carried out according to the modified protocol described by Carović-Stanko et al. [63]. Four primer combinations were used for selective amplification: VIC-EcoRI-ACG + Tru1I-CGA, NED-EcoRI-AGA + Tru1I-CGA, FAM-EcoRI-ACA + Tru1I-CGA and PET-EcoRI-ACC + Tru1I-CGA.

4.2.2. AFLP data analysis

4.2.2.1. Within-population diversity

To construct a binary matrix, the obtained AFLP fragments were scored as present (1) or absent (0). Diversity within populations was assessed by calculating the proportion of polymorphic markers (%P), the number of private markers (Npr), and the frequency down-weighted marker values (DW) [64] using the AFLPdat R package [65]. The Shannon information index of each population was calculated as I = -Σ (pi log2 pi), where pi is the phenotypic frequency [66,67]. In addition, genetic diversity (HE) was calculated using a Bayesian approach [68], assuming Hardy-Weinberg equilibrium due to outcrossing (FIS = 0) as implemented in AFLP-Surv v. 1.085 [69]. The overall mismatch error rate for all AFLP primer combinations was 2.5%. 

4.2.2.2. Population differentiation and structure

Using the pairwise distance matrix based on the Dice coefficient [70], a neighbour-joining tree was constructed and bootstrapped using 1000 replicates [71] using PAST v2.01 [72]. 

Analysis of molecular variance (AMOVA) [73] was used to partition the total genetic variance among and within populations of each taxon and between closely related taxa, among populations within taxa and within populations. The variance components were tested with 10,000 permutations in Arlequin ver. 3.5.2.2 [74]. Population structure was assessed using two Bayesian clustering approaches implemented in STRUCTURE v2.3.4 [75] and BAPS v6.0 [76]. In STRUCTURE, 30 runs were performed for each K by setting the number of clusters (K) from 1 to 21. Each run consisted of a burn-in period of 200,000 steps followed by 1,000,000 Monte Carlo Markov Chain (MCMC) replicates assuming an admixture model and correlated allele frequencies. The calculations were performed on the Isabella computer cluster at the University of Zagreb, University Computing Centre (SRCE). The most probable number of K was selected by calculating ΔK [77] in StructureSelector [78]. StructureSelector was also used to cluster and average the results of independent runs using the approach described by Kopelman et al. [79]. BAPS was applied for population mixture analysis without the geographic origin of the samples as an informative prior ('Clustering of Individuals') and with this prior ('Spatial Clustering of Individuals') [80]. The maximum number of clusters (K) was set to 20, and each run was replicated ten times. Population admixture analysis [81] was performed with the default settings.

4.3. Gas chromatography and mass spectrometry (GC-MS) analyses

The EO of each Micromeria and Clinopodium sample obtained by hydrodistillation were collected for each sample in the pentane/diethileter mixture and analyzed by gas chromatography and mass spectrometry (GC-MS). GC was performed using a gas chromatograph (model 3900; Varian Inc., Lake Forest, CA, USA) and a mass spectrometer (model 2100T; Varian Inc.). The MS conditions were: ion source temperature 200 °C, ionization voltage 70 eV; mass scan range: 40–350 mass units. The carrier gas was helium. Two columns were used: nonpolar VF-5ms and polar capillary columns CP Wax 52. The conditions for the VF-5ms column were: temperature 60 °C isothermal for 3 min, then increased to 246 °C at a rate of 3 °C min-1, and held isothermal for 25 min. The CP Wax 52 column conditions were: temperature 70 °C isothermal for 5 min, then increased to 240 °C at a rate of 3 °C min-1, and held isothermal for 25 min. The injection volume was 2 μL, and the split ratio was 1:20. The triplicate analyses of the individual peaks were identified by comparing the retention indices of the n-alkanes with literature data and authentic standards [82,83].

4.4. Principal component analysis

Principal component analysis (PCA) was based on 21 significant constituents of the essential oil. PCA was performed using the PRINCOMP procedure in SAS v9.3 [84] and the biplot showing the populations, and oil constituents (as vectors) was constructed using the first two principal components.

4.5. Mantel test

The Mantel test [85] was used to test the correlation between genetic and biochemical data matrices. Pairwise genetic distances between populations were calculated using Nei's standard genetic distance (DNei) in AFLP-Surv v1.085 [69]. Biochemical differences were calculated as Euclidean distances between populations for the first two principal components of the principal component analysis (PCA) of EO constituents. The significance level was assessed after 10,000 permutations in NTSYS-pc v2.21L [86]. 

Conclusions: please consider replacing it with

5. Conclusions

STRUCTURE analysis based on AFLP genetic data grouped the studied ten Micromeria and six closely related Clinopodium taxa distributed in the Balkan Peninsula into three genetic groups. The first cluster included all Clinopodium taxa, while Micromeria species were divided into two clusters. In general, their geographic distributions did not support the genetic grouping of the studied taxa and populations. Numerous populations of genetically well-supported taxa were also found to be scattered over large areas with no evidence of spatial groupings. Such a result suggests that substantial gene flow barriers exist between closely related taxa, precluding any possibility for inter-species hybridization and consequent fusion of these taxa into spatially structured clusters. An exception is the complex of M. juliana, M. kerneri, and M. microphylla, where these barriers are weakestGenerally, groups of taxa were much less supported than individual taxa, indicating their concurrent dispersal and approximately the same time of origin. The results also showed that the current taxonomic position of some species needs to be satisfactorily resolved. Within the C. dalmaticum group, two well-supported taxa were identified, one formerly known as M. dalmatica (today C. dalmaticum) and another as M. bulgarica (today also C. dalmaticum). The species M. graeca ssp. fruticulosa (formerly M. fruticulosa) emerged as a well-supported taxon and not a representative of M. graeca. Two populations of M. croatica, formerly recognized as M. pseudocroatica, were also clearly differentiated from any other taxon. Within the M. cristata group, taxon M. cristata ssp. kosaninii (formerly recognized as M. kosaniniilacks the needed support for its recognition as either species or subspecies. Although further studies are needed on some species within the genera Clinopodium and Micromeria, the AFLP data obtained in this research provide a good starting point for future studies. Such a study should include nuclear and plastid DNA sequences on a more significant number of populations and individuals. Finally, the Mantel test showed a weak but significant correlation between phytochemical and AFLP data.

Scientific comments:

Line 44: please further explain “70/20”. Please consider to be more precise.

Line 86: why were AFLP (and not more recent techniques) selected for molecular studies?

Line 92: please further explain “private”.

Lines 148 – 149: please further explain “The detected genetic clusters were not geographically defined at all, as representatives of each cluster were found throughout the study.”. The sentence is confusing.

Author Response

Dario Kremer

University of Zagreb

Faculty of Pharmacy and Biochemistry

A. Kovačića 1,

10000 Zagreb

Croatia

e-mail: kremer@pharma.hr

Tel.: + 38-51-4619-422

Zagreb, November, 30. 2022

            Dear Reviewer,

  First of all, the authors would like to thank the reviewer for the exceptional effort he invested in the review and proofreading of the paper. Please find enclosed answers on your comments on manuscript 'Phytochemicals and their correlation with molecular data in Micromeria and Clinopodium (Lamiaceae)’ taxa after review.

List of questions and comments are given below.

Comment: In the reviewer's opinion, the authors of this paper do not have English as their native tongue. A high degree of editing and revising will be necessary. Also, some aspects concerning assertiveness (e.g., avoid expressions such as “could be explained”), scientific accuracy (e.g., weak, but significant correlation – if the correlation is not significant, it should not be reported; please consider removing “but significant”) have to be improved, and text clarity.Due to the extension of the corrections needed, instead of presenting the suggestion by line, the reviewer presents the bulk sections of the text (suggestions in bold).

Authors' response: The comment is accepted and the manuscript has been proofread by a native English speaker. 

Comment: The Introduction section is quite confusing and should be improved for the simplification and clarity of the text.

Authors' response: The suggestions made by the reviewer have been accepted and incorporated in the Introduction section, which has also been further revised.

Comment: The Result section is deeply confusing and should be improved for the simplification and clarity of the text.

Authors' response: The suggestions made by reviewer have been accepted and incorporated in the Result section, which has been further revised. 

Comment: The Discussion section is deeply confusing and should be improved for the simplification and clarity of the text.

Authors' response: The suggestions made by the reviewer have been accepted and incorporated in the Discussion section, which has been further revised. 

Comment: The Material and Methods needs to be simplified. If the ordinary protocols were used, there is no need to describe them. It is enough to cite the reference and point out if modifications were made.

Authors' response: The suggestions made by the reviewer have been were accepted and incorporated in the Material and Methods section. 

Comment: The Conclusion section covers general concepts. An effort should be made to present conclusion related to the presented work.

Authors' response: The suggestions made by the reviewer have been accepted and incorporated in the Conclusion section, which has been further revised.

Comment: Line 44: please further explain “70/20”. Please consider to be more precise.

Authors' response: The suggestion was accepted and “70/20” has been changed into “70 or only 20”

Comment: Line 86: why were AFLP (and not more recent techniques) selected for molecular studies?

Authors' response: Both AFLP and different NGS approaches are SNP-based techniques, differing mostly in the number of detected mutations (i.e., resolution). Since this research was not oriented towards being an in-depth and highly detailed analysis of the phylogenetic relationships among the studied taxa, but rather a comparative analysis of the genetic and phytochemical relationships among the studied taxa, we selected AFLP over other, more demanding NGS approaches. We do not believe that the results would have given major differences with the use of a higher-resolution approach.

Comment: Line 92: please further explain “private”

Authors' response: The private allele is unique for the specific population, and it cannot be found in any other studied populations. The suggestion was accepted and the explanation was added in this sentence: ''Of the 1694 polymorphic markers in 434 individuals, 19 were private (unique to a specific population).''

Comment: Lines 148 – 149: please further explain “The detected genetic clusters were not geographically defined at all, as representatives of each cluster were found throughout the study.” The sentence is confusing.

Authors' response: We agree with the Reviewer and have revised this sentence to read: “The genetic clusters were not geographically defined, since representatives of each cluster were found throughout the studied area.”

  Sincerely, 

Dario Kremer
